# Stereoselective gridization and polygridization with centrosymmetric molecular packing

Dongqing Lin[1,4], Ying Wei[1,4], Aizhong Peng[1], He Zhang[1], Chunxiao Zhong[1], Dan Lu[2], Hao Zhang[2], Xiangping Zheng[1], Lei Yang[1], Quanyou Feng [1], Linghai Xie [1,3 ✉] & Wei Huang [1,3 ✉]

The gridarenes, with well-defined edges and vertices, represent versatile nanoscale building blocks for the installation of frameworks and architectures but suffer from difficulty in stereoselective control during their synthesis. Here we report a diastereoselective gridization of superelectrophilic diazafluorene-containing substrates ($A_mB_n$) with crescent shapes into Drawing Hands grids (DHGs). The *meso*-selectivity reaches 75.6% diastereomeric excess (*de*) during the gridization of $A_1B_1$-type substrates and maintains ~80% *de* during the polygridization of $A_2B_2$-type monomers. Such stereocontrol originates from the centrosymmetric molecular packing of two charge-delocalized superelectrophiles with synergistically $\pi-\pi$ stacking attractions and coulombic repulsions. As *meso*-stereoregular structures show 20~30 nm in length, the rigid ring/chain-alternating polygrids have a Mark–Houwink exponent of 1.651 and a molecular weight (*M*) dependence of the hydrodynamic radius $R_h \sim M^{1.13}$. Via the simulation of chain collapse, *meso*-configured polygridarenes still adopt rod-like conformations that facilitate the high rigidity of organic nanopolymers, distinguished from toroid backbones of *rac*-type polygrids.

[1] Centre for Molecular Systems and Organic Devices (CMSOD), Key Laboratory for Organic Electronics and Information Displays and Jiangsu Key Laboratory for Biosensors, Institute of Advanced Materials (IAM), Nanjing University of Posts and Telecommunications, 9 Wenyuan Road, Nanjing 210023, China. [2] State Key Laboratory of Supramolecular Structure and Materials, College of Chemistry, Jilin University, 2699 Qianjin Avenue, Changchun 130012, China. [3] Shaanxi Institute of Flexible Electronics (SIFE), Northwestern Polytechnical University (NPU), 127 West Youyi Road, Xi'an 710072, China. [4] These authors contributed equally: Dongqing Lin, Ying Wei. ✉email: iamlhxie@njupt.edu.cn; provost@nwpu.edu.cn

Cyclophanes are heteroatom/C($sp^3$)-hybridized arene macrocycles, which have been transformed into calix[$n$]arenes[1], pillar[$n$]arenes[2], and three-dimensional (3D) cages[3] in the application of host–guest recognition[4] since they were first synthesized in 1949[5]. Further, these structures have been interlocked into catenanes[6], rotaxanes[7], and knots[8], as the candidate for molecular machines[6]. However, most of them are difficult to construct nanoarchitectures such as nanorods, frameworks, and nanorings that have been constructed by porphyrin derivatives[9,10]. Inspired by their presence in fullerenes[11], carbon nanotubes[12], and warped nanographene[13], fluorenyl groups were introduced into nanoscaffolds that endow enhanced rigidity, orthogonal backbones, and predetermined extension sites, as well as wide bandgap properties[14] and molecular motor behaviors[15]. According to grid complexes from Lehn and colleagues[16], they are called gridarenes[17] because of well-defined edges and vertexes that afford them extendability, installability, programmability, and scalability. These nanoscale gridarenes with the highest symmetric backbones can be categorized into five types involving $C_2$-symmetric angle-lost types with two extension sites, $C_{2v}$-symmetric ladder types[18] with two pairs of parallelly distributed extension sites, $C_4$-symmetric windmill types[19] with four radial distributed extension sites, $C_2$-symmetric rhombus types with six extension sites, and $C_{2v}$-symmetric tic-tac-toe types[20] with eight extension sites. However, the C($sp^3$) atom at the 9-position of a fluorenyl moiety serves as a chiral center that results in various stereoisomers during gridization reactions[19]. For example, windmill-like gridarenes[19] have 6 stereoisomers within 2 pairs of enantiomers, ladder-like grids[18] have 7 stereoisomers within 2 pairs of enantiomers, and 日-shaped digrid nanoscaffolds[21] even have 27 stereoisomers within 12 pairs of enantiomers. The tacticity of gridarenes[18,20] and polygridarenes would potentially affect the crystallinity[22], exciton behaviors[23], and drug activity[24]. Therefore, it is significant to control stereoselective synthesis that would benefit for the molecular engineering of state-of-the-art model and materials.

Stereoselectivity arises from asymmetric attack orientations of intermediates, commonly induced by molecular orbital interactions[25], steric effect[26], electronic effect[26], and supramolecular bonds[27]. In detail, the preferred attack orientation originates from the most stable molecular arrangement[26], which can be dominated by the packing mode of substrates with sufficient noncovalent interactions including hydrogen bonds[28], π–π stacking[29], and coulombic repulsion[30]. Encouraged by this, we exploit synergistically supramolecular attraction–repulsion interactions[31] to tentatively control over the stereoselective pathways of carbocation

intermediates during the Friedel–Crafts gridizations and polygridizations (gridization-type polymerizations). To eliminate the stoichiometric effect[32] on gridizations, we design heterodifunctional $A_1B_1$-type substrates consisting of an alcohol group as the A-part and a benzenoid moiety as the B-part. For the A-part with the same fluorenyl skeleton, we select a diazafluorenyl group (DAF), which is a kind of electron acceptors[33] with metal coordination[34], superelectrophile[35], and hydrogen bonding behaviors[36]. For the B-part, a carbazole moiety (Cz) with a 3-position linked to a DAF group is used to construct mono-crescent-shaped substrates, called MCs. Distinguished from other $A_1B_1$-type substrates with a thiophene moiety[19] or a Cz moiety linked at a 1-/2-/9-position, MCs take advantage of geometry matching to form gridarenes with only two chiral centers and minimized strain energy. These gridarenes in angle-lost shape resemble the famous lithograph art of "Drawing Hands"; thus, we coin them Drawing Hands gridarenes (DHGs, in Fig. 1). During the gridization of MC substrates, we unexpectedly observe that most of DHGs exhibit *meso*-configurations rather than thermodynamic *rac*-isomers. Moreover, we demonstrate such gridization rule to transform into stereoselective polygridization, as the inverse process of the ring-opening polymerization[37], to afford rigid rod-like polygrids with *meso*-tacticity.

## Results

**Investigation of gridization rules**. Before the gridization of $A_1B_1$-type substrates, we carried out a Friedel–Crafts prototype reaction of diazafluorene-alcohol **1a** with Cz in a 1:1 equivalent ratio (Fig. 2a). Considering the formation of tricationic superelectrophiles[35,38] on the DAF moieties, we used $CF_3SO_3H$ (75 equiv) to initiate the reaction that produced monosubstituted **2a** in 14% yield and disubstituted **3a** in 58% yield but the dehydroxylated byproducts (**4a** and **5a**, in Supplementary Fig. 1) were also generated. Luckily, incorporating a methoxyl group (substrate **1b**) led to monosubstituted **2b** in 13% yield and disubstituted **3b** in 85% yield within 10 s, which reveals the in situ acceleration of the reaction with monosubstituted products. Interestingly, such acceleration was not observed in the reaction system where the fluorenyl substrate **F-1b** generated disubstituted **F-3b** in 22% yield (Supplementary Fig. 2). As electronic effects cannot explain this phenomenon, a more rational explanation is proposed based on intermolecular aggregation via hydrogen bonds between DAF moieties and strong acid molecules[39]. Considering the stronger hydrogen bonds derived from increased acidity[40], we hypothesized that this aggregation strength could be tuned by Brønsted acid additives before initiating the reaction.

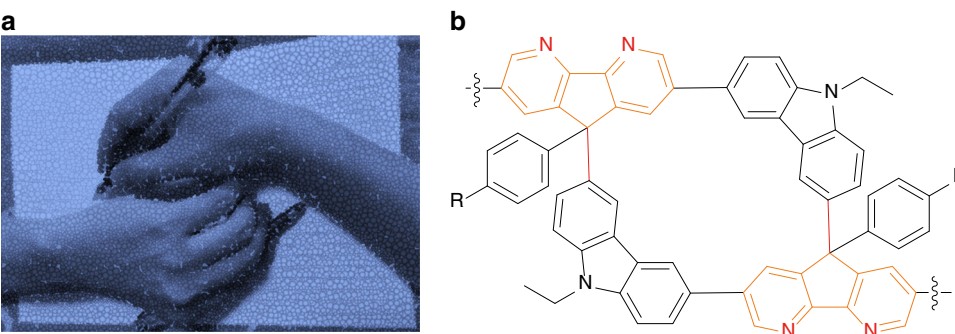

**Fig. 1 The conceptual scheme of Drawing Hands art and molecular Drawing Hands gridarene (DHG) structures as a model of the nano-linkage. a** The picture of Drawing Hands art created by one of the corresponding author Professor Linghai Xie, without using any element from a database. The ball-point pens are analogous to the C($sp^2$)-C($sp^3$) bonds between the 9-position of diazafluorenyl groups and the 3-position of the carbazole segments. **b** The molecular structures of Drawing Hands gridarenes. The orange lines and two nitrogen atoms (in red) depict the diazafluorenyl groups (DAFs). The red lines represent the linkage between the 9-position of diazafluorenyl groups and the 3-position of the carbazole segments, which occurs during the gridization process.

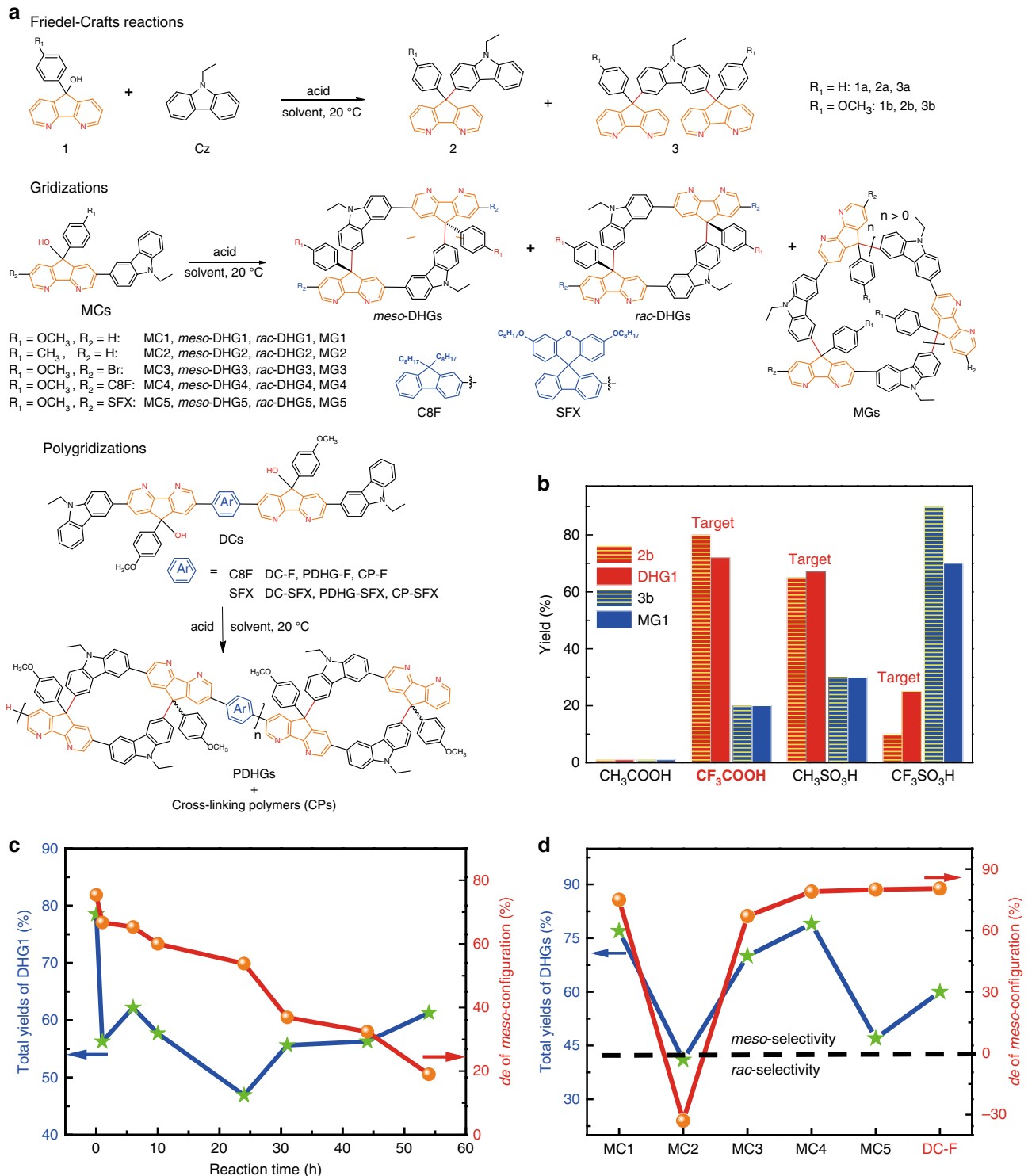

**Fig. 2 Characteristics of the Friedel–Crafts reactions, gridizations, and polygridizations. a** Corresponding formula. The $R_1$ and $R_2$ groups are depicted in red and blue, respectively. **b** The dependence of acid additives on the yields (after 30 s of reactions). The volume ratio of the acid additive to dichloromethane (DCM) is about 1/3. The concentration of substrates is about 10~15 mM. For the Friedel–Crafts reactions, the height of red and yellow column corresponds to the yield of **2b**; the height of blue and yellow column represents the yield of **3b**. For the gridization reactions, the heights of the pure red column and the pure blue column correspond to the yields of DHG1 and MG1, respectively. **c** The dependence of gridization times on the *meso*-selectivity of MC1, under the conditions of DCM solvent, $CF_3COOH$ acid additive, and $CF_3SO_3H$ acid catalyst. The blue line and green stars exhibit the total yields of DHG1, and the red line and orange dots show the *de* value of *meso*-configuration. **d** The total yields (green dot and blue line) and the *meso*-selectivity of DHG-based products (orange dot and red line) under the optimized conditions in 1 min. The black dashed line is the boundary between the *meso*-selectivity and *rac*-selectivity. The yields of PDHG-F is defined as the sample after fully undergoing Soxhlet extraction.

Consequently, we observed the dependence of **3b** yield on the acidity of the acid additives (Fig. 2b), including $CH_3SO_3H$ (35%), $CF_3COOH$ (20%), and $CH_3COOH$ (0%). These results suggest that the reaction pathways could be flexibly tuned by intermolecular aggregation (reconfirmed by the solvent dependence in Supplementary Fig. 3).

Based on prototype reactions, we investigated the gridization (in Fig. 2a) of substrate MC1 with a methoxyl group that eliminates the generation of dehydroxylated byproducts. However, we observed that the conditions favoring disubstitution ($CF_3SO_3H$ additive, dichloromethane (DCM) solvent) generate target DHG1 in only 8% yield, whereas they afford macrogridarene oligomers MG1 in 42% yield. This results suggest that the serious aggregation are unfavorable for the formation of DHG products. In contrast, the moderated aggregation conditions (favoring monosubstitution) afford DHG1 (within 1 min) in 68% yield via the $CH_3SO_3H$ additive and in 77% yield via the $CF_3COOH$ additive (Fig. 2b). More unexpectedly, under the condition of $CF_3COOH$ additive, we found that the ratio of *meso*-DHG1:*rac*-DHG1 was 7.2:1, which is identical to 75.6% diastereomeric excess (*de*) of the *meso*-configuration. Considering the thermodynamic equilibrium of Friedel–Crafts reactions[41], we prolonged the reaction time and found that the total DHG1 yields were overall reduced from 77% to 55% (Fig. 2c). Further, the *meso*-stereoselectivity was reduced from 75.6% *de* (1 min), 53.8% *de* (24 h) to 19.0% *de* (54 h), corresponding to the gradually lowered *meso*-DHG1 yields from 69% to 36.5% (Supplementary Fig. 4). These results indicate that the *meso*-DHG1 is the kinetic product that can transform into MG1. Thus, ultrafast quenching (1 min after the gridization) is key to the *meso*-selective gridization. However, when using methyl-based MC2 substrates, the *meso*-selectivity can transform into *rac*-selectivity (19 ~ 26% *de* of *rac*-DHG2) along with negligible gridization time effect (Supplementary Figs. 5 and 6). Hence, the oxygen of a methoxyl group plays a key role in dominating the kineic *meso*-selective pathways. To further test the scope of diastereoselectivity, MC1 derivatives with various $R_2$ groups were reacted under the optimized conditions (Fig. 2d). For example, the Br-containing substrate MC3 affords *meso*-DHG3 in 61% yield and 67% *de*, indicating that the *meso*-selectivity cannot be evidently affected by bromine atoms. Furthermore, the *meso*-stereocontrol is maintained if incorporating 9′9-dioctylfluorenyl group C8F (MC4, in 82% *de*) or spiro(fluorene-9,9′-xanthene) group SFX (MC5, in 80% *de*), although the SFX group decreases the DHG5 yield to 43%. Overall, $R_2$ groups can affect DHG yields but seemingly do not destroy the *meso*-selectivity.

Inspired by the above observations, we intermediately performed polygridization (in Fig. 2a) of $A_2B_2$-type dicrescent-shaped monomers (called DCs). To efficiently obtain the target polymers of Drawing Hands gridarenes (called PDHGs) instead of insoluble cross-linked polymers (CPs), we selected the conditions favoring monosubstitution ($CF_3COOH$ additive, DCM solvent). Unfortunately, the DC-F monomer still afforded the CP-F in at least 80% yields (1 min after the reaction), even if the DC-F concentration (1 ~ 100 mM) was further optimized. These results probably originate from ring-chain competition kinetics between the formation of intermolecular C–C bonds and the pathways of intramolecular gridization[42]. However, we observed that CP-F was not generated until 40 s of reaction. As a result, 15 ~ 25 s after the initiation, we added MC1 to terminate the active chain ends (followed by quenching in 1 min) and fortunately obtained PDHG-F (in ~60% yields after Soxhlet extraction) without CP-F (Supplementary Fig. 7). Further, even if terminating in 20 s, the slow quenching (15 min after the polygridization) still efficiently generated insoluble CP-F, which is related to thermodynamic equilibrium toward a ring-opening

process (Supplementary Fig. 8). Consequently, quick termination and quenching is crucial to the polygridization. For the PDHG-F backbones, the ratio of *meso*:*rac* was calculated to be 8.8:1 via the hydrogen nuclear magnetic resonance ($^1H$ NMR) spectra (see below), corresponding to 79.5% *de* of the *meso*-selectivity (Fig. 2d). Such stereocontrol efficiency, consistent with the gridization time effect of MC1 (<1 min), is almost maintained when altering reaction times (6~25 s) and DC-F concentrations (15~60 mM, in Supplementary Fig. 9). In addition, the other polygridization from the monomer DC-SFX still afforded the *meso*-selective polygrids PDHG-SFX (~50% yields after Soxhlet extraction) but the formation of cross-linked byproduct CP-SFX was unavoidable in at least ~30% yields, as similar to the effect of the SFX group on the gridization yields.

**Characterizations on the backbone tacticity.** To demonstrate the structures of DHG products, we grew single crystals via slow evaporation from methanol/DCM mixtures. These crystals reveal that each DHG backbone possesses an inner pore consisting of 18 $sp^2$-carbon atoms and 2 $sp^3$-carbon atoms in a centrosymmetric distribution (Fig. 3a). The two chiral quaternary carbon atoms at the 9-site of DAF moieties serve as the gridarene vertices. The conjugated linkage between the DAF (at the 7-site) and Cz moieties forms the gridarene edges. Each edge adopts the *anti*-conformation where two 9-sites of the DAF and Cz moieties are oriented in opposite directions. In detail, *meso*-DHG1 and *rac*-DHG1 are differentiated by their crystal cells and geometric backbones. For *meso*-DHG1 with the space group P-1, it shows a $C_2$-symmetric backbone with a geometric size of $1.34 \times 0.98$ $nm^2$. By contrast, with the space group C12/c1, *rac*-DHG1 exhibits an asymmetric folded scaffold with a geometric size of $1.35 \times 0.91$ $nm^2$. For the two scaffolds, *meso*-DHG1 has dihedral angles of 129.0° on the vertices and 43.8° on the edges, whereas *rac*-DHG1 exhibits dihedral angles of 108.0° on the vertices and 14.2° on the edges.

The backbone differences further discriminate the infrared vibrational transitions between *meso*- and *rac*-configurations, which is shown in the Fourier transform infrared spectra (FT-IR) in Supplementary Fig. 10. For the 1024 $cm^{-1}$ (C–O stretching vibrations) and the 1100 ~ 1080 $cm^{-1}$ (C–H scissoring vibration on the methoxybenzenes) bands, the folded *rac*-DHG1 exhibits much stronger intensities than those of $C_2$-symmetric *meso*-DHG1 whose centrosymmetrically vibrational distribution causes little changes in dipole moment (Supplementary Fig. 11). For asymmetrically vibrational modes, the centrosymmetric backbone of *meso*-DHG1 enhances infrared vibrational activities both at the 1248 ~ 1264 $cm^{-1}$ (asymmetric C–O stretching vibrations) and the 1294 $cm^{-1}$ (asymmetric C–H scissoring vibration on the DAF groups) bands vs. the asymmetric configuration of *rac*-DHG1. More interestingly, the band intensity ratio of 1294 $cm^{-1}$ to 1024 $cm^{-1}$ ($I_{1294}/I_{1024}$) is gradually enhanced from 0.13 to 0.76 when increasing the *meso*-DHG1 proportion from 0 to 100% in the mixing systems (Fig. 3b and Supplementary Fig. 12). Meanwhile, the band intensity ratio of 1248 ~ 1264 $cm^{-1}$ to 1024 $cm^{-1}$ ($I_{1248}/I_{1024}$) is continuously increased from 0.56 to 1.55. The above vibrational features of *meso*-DHG1 are also observed in PDHG-F that exhibits $I_{1294}/I_{1024} = 0.66$ and $I_{1248}/I_{1024} = 1.40$, which probably reveals 80% *de* of *meso*-configurations.

The structural differences between DHG diastereomers were further demonstrated by the shielding/deshielding effect of aromatic ring currents in $^1H$ NMR spectra (Fig. 3c and Supplementary Figs. 13–18). For *meso*-DHG1 and *meso*-DHG4, the ring currents of DAF moieties exclusively shield the *a* and *b* protons on the methoxybenzyl groups, which is evidenced by their chemical shifts (6.95 ~ 6.65 parts per million (p.p.m.)) being smaller than

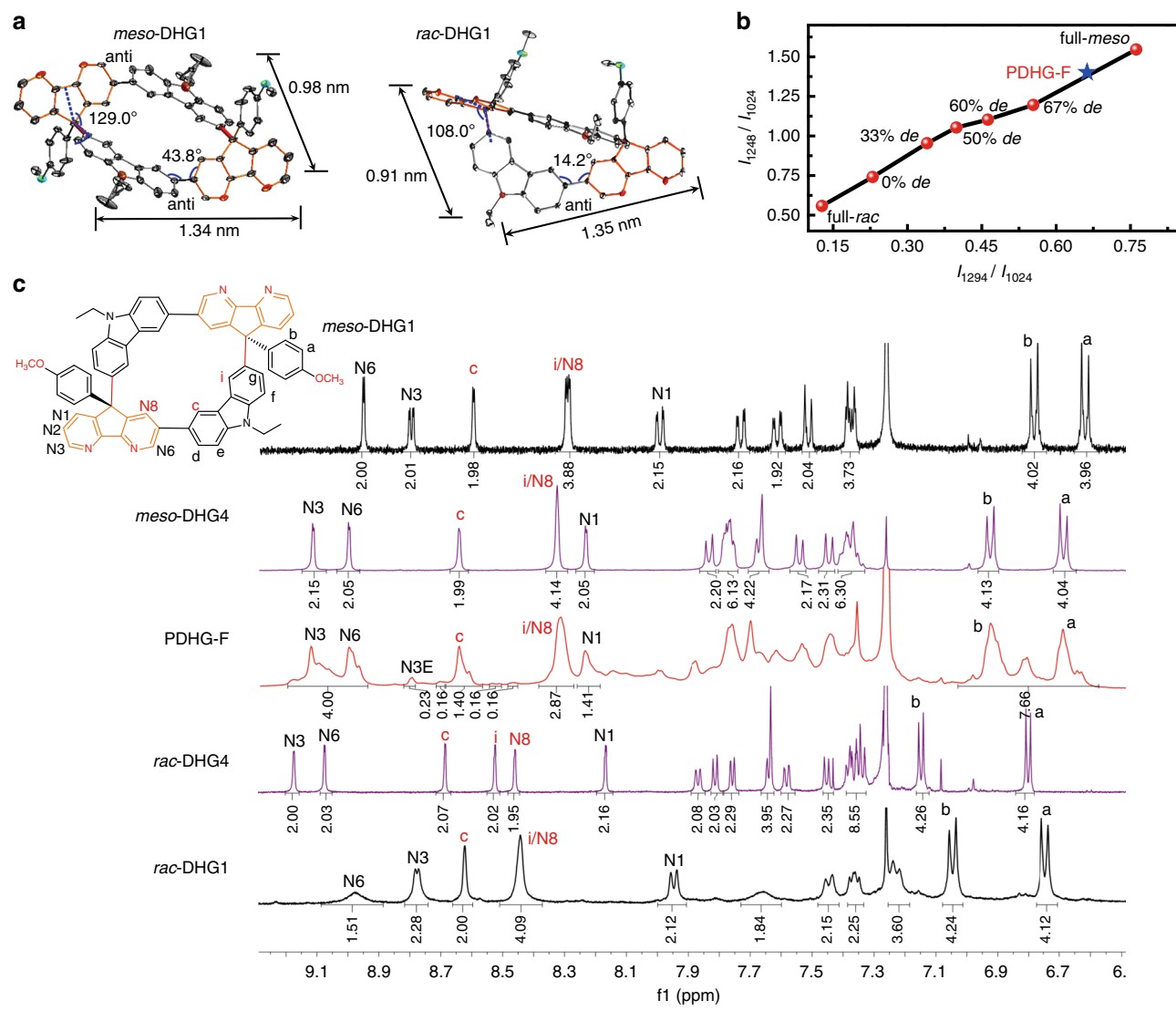

**Fig. 3 Characterizations of the tacticity of DHGs and PDHG-F backbones. a** The single-crystal crystallography of *meso*-DHG1 and *rac*-DHG1s. The blue lines exhibit the dihedral angles of the backbones. **b** The integral ratio of the vibrational absorption at 1248 ~ 1264 cm⁻¹ band to 1024 cm⁻¹ band ($I_{1248}/I_{1024}$) and the intensity ratio of the 1294 cm⁻¹ band to 1024 cm⁻¹ band ($I_{1294}/I_{1024}$) for the specific mixing systems (*meso*-DHG1 and *rac*-DHG1). These results are extracted from the Fourier transform infrared spectra of mixing samples with well-defined doping percentage (transformed to *de* value of *meso*-selectivity). For example, 0% *de* = 1:1 of *meso*-DHG1:*rac*-DHG1; full-*rac* represents the pure *rac*-DHG1 sample; full-*meso* represents the pure *meso*-DHG1 sample. The red dots exhibit the band intensity ratio of DHG1 mixing systems; the blue star shows the band intensity ratio of PDHG-F. **c** The ¹H NMR spectra of DHG1 (in black lines, as the end-capping units), DHG4 (in purple lines, as the repeat units), and PDHG-F (in red lines). The assignments of *meso*-DHG4 were made in reference to the ¹H–¹H correlation spectroscopy results. The key positions associated with the stereoselective calculation (such as c, i, and N8 sites) are marked in red.

those in *rac*-DHG1 and *rac*-DHG4 (7.15 ~ 6.75 p.p.m.). For both *rac*-DHGs, the folded aromatic scaffold enhances the deshielding effect on the protons of the inner pores, displaying a chemical shift in the *i* and N8 protons (8.55 ~ 8.40 p.p.m.) larger than that in both *meso*-DHGs (8.40 ~ 8.30 p.p.m.). The above observations will be applied in PDHG-F if their chain conformations are expanded rather than collapsed, because chain collapse induces compact intramolecular stacking, which upshifts overall proton signals[6]. According to the results of the molecular dynamic simulation, we evaluated the expansion factor β > 1 for PDHG-F chains in CHCl₃ solvent (detailed in Supplementary Note 4), indicating the expanded conformations. Such evaluation is confirmed by the overall proton signals of PDHG-F with approximately the same chemical shifts as DHG4. Thus, the *c*, *i*/N8 protons, with discriminated chemical shifts between the *meso*- and *rac*-configurational repeat units, can be used to calculate the *de* value of *meso*-selectivity via the integral

ratio. For the *c* sites of PDHG-F, the integral ratio in 8.65 p.p.m. (consistent with *meso*-DHG4, an integration of 1.40) to 8.70 p.p.m. (identical to *rac*-DHG4, an integration of 0.16) displays the 79.5% *de* of the *meso*-selectivity. For the *i*/N8 sites of PDHG-F, the total integral ratio in 8.32 p.p.m. (consistent with *meso*-DHG4, the total integrations of 2.87) to 8.54 ~ 8.48 p.p.m. (identical to *rac*-DHG4, the total integrations of 0.32) evaluates the 79.9% *de* of *meso*-configuration. Moreover, these *meso*-selective results of PDHG-F are reconfirmed by their chemical shift distribution of *a*, *b* signals that are dominantly assigned to *meso*-DHGs, according to the ¹H–¹H correlation spectroscopy (Supplementary Fig. 18).

**Characterizations on the linear PDHG-F backbone.** Given the absence of a branched scaffold, we hypothesized that PDHG-F would possess rod-like chain properties. In this regard, its

Mark–Houwink exponent $\alpha$ would be ~1.7 instead of 0.3 ~ 0.5 (for branched backbones[43])[44]. The Mark–Houwink exponent of PDHG-F ($\alpha_{LP}$) can be deduced by its gel permeation chromatography (GPC) calibration curve that shows a linear relationship between logarithmic molecular mass (lg $M$) and elution time ($t$)[9]. Its curve slope ($k_{LP}$) can transform into $\alpha_{LP}$ via the equation (detailed in Supplementary Note 5): $(\alpha_{LP} + 1)/(\alpha_{PS} + 1) = k_{PS}/k_{LP}$, based on the known values including the slope of the polystyrene (PS) calibration curve ($k_{PS} = -0.198$, during the elution time of 18 ~ 23 min) and the Mark–Houwink exponent of PS ($\alpha_{PS} = 0.714$)[45]. To obtain $k_{LP}$, we analysed the chromatograph of PDHG-F with an average lower molecular weight and tentatively assigned their elution peaks (19 ~ 23.4 min, all in ~1.01 polydispersity index) to corresponding degree of polymerization (DP) of 2 ~ 6 (Fig. 4a). In this case, we established a calibration equation lg $M = -0.128\ t + 6.329$ where $k_{LP} = -0.128$ transforms into $\alpha_{LP} = 1.651$. Therefore, such high $\alpha_{LP}$ reconfirms our hypothesis of the rod-like main chain for PDHG-F backbones (up to DP = 6).

Deeply, we demonstrated the PDHG-F scaffolds with higher DP via calculating the number of chain ends ($N_{end}$). Theoretically, the linear scaffold displays $N_{end} = 2$, whereas the branched scaffold shows $N_{end} \geq 3$. $N_{end}$ can be evaluated[6] through the equation $N_{end} = (DP \times 4I_{N3E})/(I_{N3+N6} + I_{N3E})$, where $I_{N3E}$ is the integration of the chain-end N3E proton signals at 8.80 p.p.m. and $I_{N3+N6}$ is the total integrations of the N6 and N3 protons at 9.15 ~ 8.95 p.p.m. (in Fig. 3c). To obtain the average DP, we divided the elution peak (15 ~ 22.5 min) in the light of specific elution times corresponding to individually well-defined DP (secondary elution peaks in Fig. 4b), based on the extrapolated calibration curve of the PDHG-F system (Supplementary Table 3). Such division can obtain the fraction of each specific DP, which transforms into the height of red stars in Fig. 4b. Further, via discretely integrating each DP value in individual fraction (according to equations ES12 and ES13 in Supplementary Note 5), we calculated the number- and weight-average degrees of polymerization $DP_n = 8.5$ and $DP_w = 9.5$, respectively. DP = 8.5 transforms into $N_{end} = 1.85$, which indicates the linear chains with a maximum DP of 12 (~30 nm in length, 35.4% proportion).

Moreover, a molecular weight dependence of hydrodynamic radius ($R_h$) was tested to demonstrate the rigidity of PDHG-F ($R_h \sim M^1$ for stretched chains) via dynamic light scattering (DLS). According to the size distribution (Supplementary Fig. 21), we obtained their $R_h = 11.1 \sim 14.2$ nm ($DP_w = 9.1 \sim 10.2$), which represents the radius of the hydrodynamic sphere model (Fig. 4c). As these sizes are slightly longer than the half of a PDHG-F main chain, we deduced that this spherical diameter is similar to a PDHG-F backbone. Further, the scaling law of

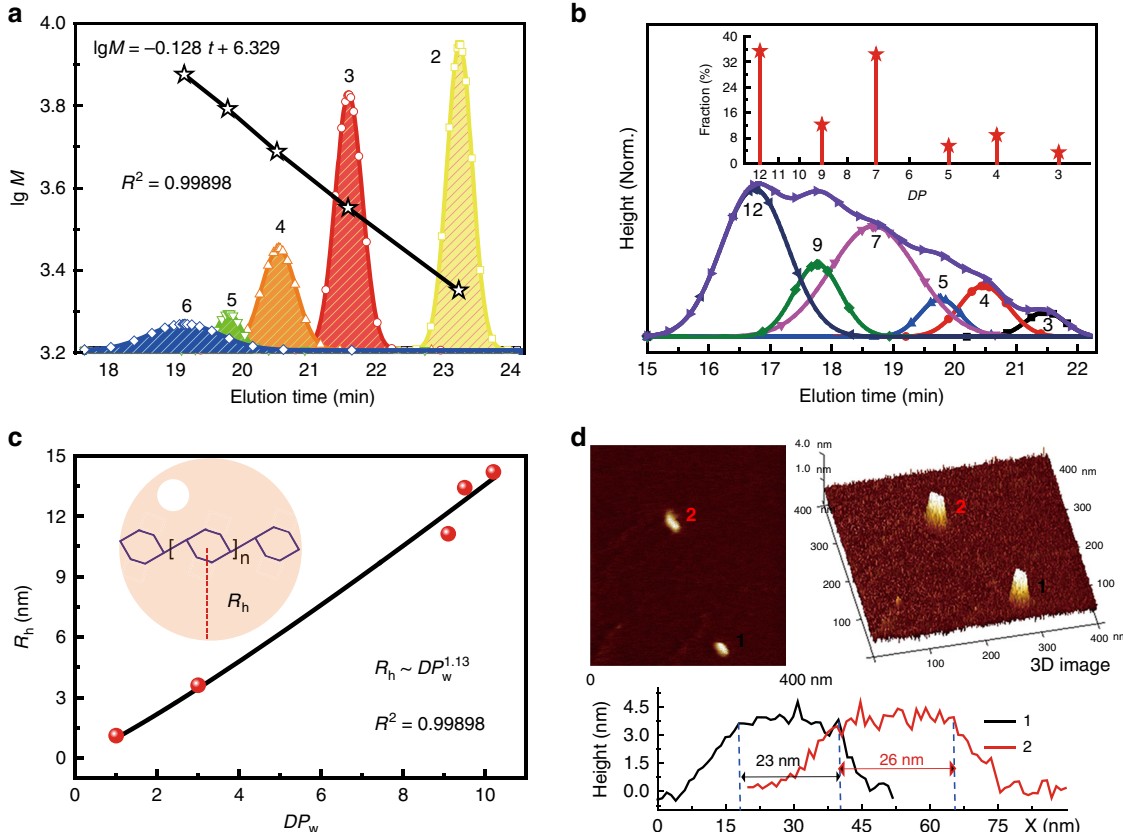

**Fig. 4 Characterizations of the linear configuration of PDHG-F main chains. a** The calibration of PDHG-F with lower molecular weight (the coefficient of determination $R^2$ is 0.99898). The yellow, red, orange, green, and blue peaks corresponds to dimer, trimer, tetramer, pentamer, and hexamer, respectively (in which the star dots exhibit their individual logarithmic molecular mass lg $M$ and elution time $t$). **b** The GPC analysis of PDHG-F. The broad elution peak (violet lines) is divided into a series of gauss peaks (in black, red, blue, magenta, olive, and navy lines) with well-defined degree of polymerization, in terms of the extrapolation of PDHG-F calibration. The height of red star columns represents the proportion of each degree of polymerization. **c** The hydrodynamic radius ($R_h$) of PDHG-F with different weight-average degrees of polymerization (DPs), marked in red dots. The black lines exhibit the DP dependence of hydrodynamic radius $R_h$ ($R^2 = 0.98473$). The blue lines depict the linear PDHG-F backbones. Its hydrodynamic model in solutions is represented as the orange sphere whose radius (red line) is identical to $R_h$. **d** The atomic force microscope images of the PDHG-F chains (two chains labeling 1 and 2) including 2D and 3D image. In the height analysis, the size between two blue dashed lines (above 3.2 nm) exhibits the chain length of visualized PDHG-F (black for 1 and red for 2).

$R_h \sim DP_w^{1.13} \sim M^{1.13}$ approximately exhibits the linear chain conformation with high rigidity (up to ~28 nm). Such rod-like backbones are further visualized by atomic force microscopy. In Fig. 4d, two PDHG-F chains individually display 23 nm (labeling 1) and 26 nm (labeling 2) in length, which are approximate to the DLS results.

**Gridization mechanism.** Based on the above experimental results, we performed mechanistic studies on the gridization process between two MC1 substrates. In particular, via nuclear Overhauser effect spectroscopy (NOESY, in Supplementary Fig. 22a), the protonated MC1 substrates were analysed to adopt both *anti-* and *syn-*conformations in solution. Further, the formation of DHG1 results from three conformational combinations (Fig. 5a) as follows: (i) two *syn-*conformations (I, II); (ii) mixed *anti-* and *syn-*conformations (III, IV); and (iii) two *anti-*conformations (V, VI). Among of them, overall *rac-*configurations (II, IV, and VI for 12.0, 12.4, and 0 kcal mol$^{-1}$, respectively) are more stable than *meso-*configurations (I, III, and V for 90.3, 18.9, and 4.9 kcal mol$^{-1}$, respectively). In this regard, the thermodynamic pathway is not the rate-determining step during the *meso-*selective gridization.

Furthermore, considering the aggregation effect on the gridization pathways, we deduced that the formation of DHG1 is possibly related to the process of dimer packing between two MC1 molecules. Given the donor–acceptor packing mode within multicationic repulsion[39], we hypothesized that two MC1 superelectrophiles would adopt an antiparallel stacking geometry in face-to-face orientation between a DAF group and a Cz group. Such hypothesis is confirmed by the space correlations in the NOESY spectrum (Supplementary Fig. 22b). Moreover, the PDHG-F products generated from polygridizations also reconfirm the antiparallel stacking process. According to the orientation of two methoxybenzyl groups (Fig. 5b), the antiparallel stacking can be categorized into the centrosymmetric geometry VII (in the opposite orientations that generate *meso-*DHG1) and the asymmetric geometry VIII (in the same orientation that obtains *rac-*DHG1). In detail, the centrosymmetric stacking orients two DAF groups in the opposite directions and thus suffers from less intermolecular coulombic repulsion[39] (especially at the 4,5-sites of DAF groups) than the asymmetric stacking, which is similar to other pyridinium derivatives[29,30]. More significantly, the intramolecularly tricationic repulsion induces charge delocalization[46] that transforms the methoxybenzyl moiety into the planarized quinonemethide-type structure[38], which further strengthens the antiparallel and centrosymmetric packing for *meso-*selectivity (Supplementary Fig. 23). Such transformation is confirmed by the *rac-*selectivity from tolyl-based MC2 substrates in the absence of quinonemethide-type structures. Regarding the *meso-*stereocontrol results, we deduced that the more stably antiparallel stacking could dominate the reaction pathways that merely generate I, III, and V conformations. However, the single-crystal crystallography of *meso-*DHG1 and *rac-*DHG1 demonstrate the formation of V (major products) and VI (minor products), whereas both I and III with much higher conformational energies were not observed. In all, the powerfully centrosymmetric packing results in the *meso-*selectivity, under the prerequisite conditions of relatively stable products' conformations.

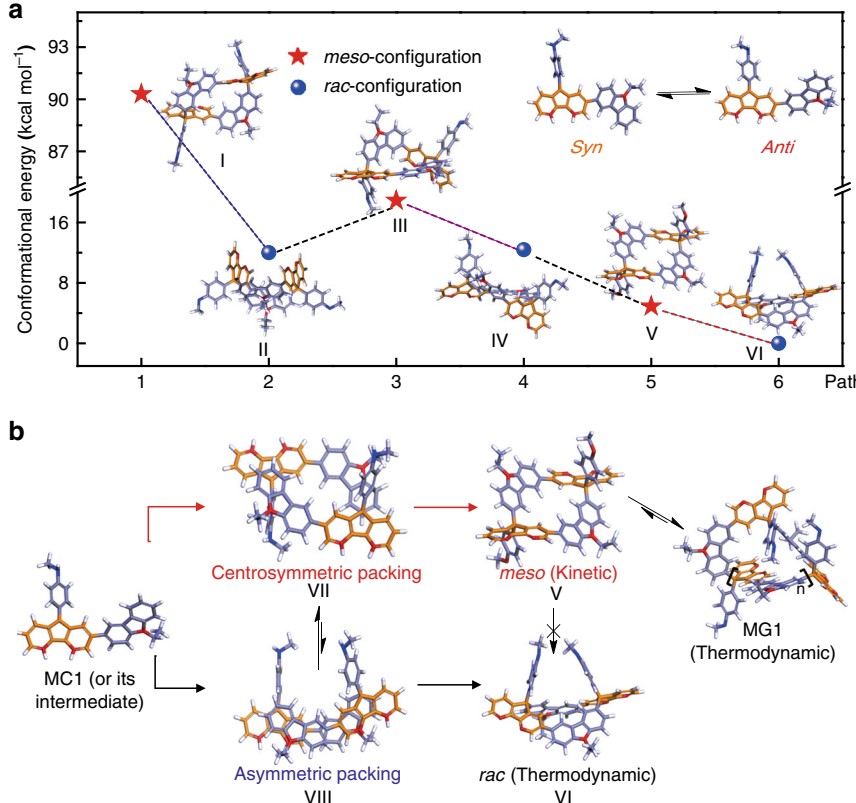

**Fig. 5 The plausible gridization mechanisms. a** The relative energies of the DHG1 products with different conformations (from individual reaction pathways). The relative energy of *anti-anti-*conformational *rac-*DHG1 is defined as 0 kcal mol$^{-1}$. Both I and II consist of two *syn-*conformational MC1 substrates; both III and IV consist of an *anti-*conformational MC1 and a *syn-*conformational MC1; both V and VI are composed of two *anti-*conformational MC1 substrates. The red stars and blue beads represent the *meso-*configurations and the *rac-*backbones, respectively. **b** The gridization mechanism of MC1.

Taken together, the above results reveal that the real gridization pathway should be determined by the competition between the kinetic stacking stage and the thermodynamic bonding stage. The strong noncovalent interactions enhanced by quinonemethide-type structures form the stable antiparallel and centrosymmetric assembly VII, which overcomes the thermodynamic barrier to dominantly generate *meso*-DHG1. Further, *meso*-DHG1 transforms into MG1 during which the thermodynamic control is favorable to the ring-opening process toward the formation of MG1 (Fig. 5b). On the contrary, due to the absence of strong self-assembly (confirmed by NOESY that does not display corresponding cross peaks, in Supplementary Fig. 24) and quinonemethide-type backbones, the methyl-based MC2 substrates are subjected to the thermodynamic control that facilitates the *rac*-selectivity and random oligomerization pathways.

The DC-F monomers are similarly controlled by the above packing mode, which not only maintains the *meso*-selectivity but also results in the abnormal polymerization kinetics (contradictory to the second-order rate rules[42], shown in Supplementary Figs. 25 and 26). According to primary results, enhancing the DC-F concentration from 15 to 60 mM leads to decreasing the $DP_n$ from 9.2 to 8.1 rather than increasing in proportional manners; prolonging the polygridization time (before termination) from 6 to 25 s slightly increases the $DP_n$ from 8.4 to 9.2, without proportional modes as well. These results indicate that the supramolecular packing enables to consistently change reaction rate constant in situ.

**Simulations of chain collapse in different tacticity**. Inspired by the analogous tacticity in polyesters[37] and poly(vinylether)s[22], we study the effect of *meso*-configurations on the chain collapse via atomistic simulation. As the enthalpic driving force[47] is crucial to the collapse of stiff backbones[48], both PDHG-F chains (full *meso*-configurational *meso*-PDHG-F and full *rac*-configurational *rac*-PDHG-F individually) are simulated in vacuum that provides dominantly intramolecular segment–segment interactions[49]. During the dynamic equilibrium (Constant volume/constant temperature ensemble (NVT), 298 K), *meso*-PDHG-F performs the conformational transition from the linear rod (expanded chains (EC stage)), via the folded sausage (sausage-like chains (SC stage)) to the collapsed rod (cluster stage, in Fig. 6a). The eccentricity of its ellipsoid model is relatively decreased from 0.99 to 0.91 (Fig. 6b), which still exhibits highly structural anisotropy that is similar to the main chains of one-dimensional nanopolymers[50]. In contrast, *rac*-PDHG-F undergoes the conformational transition from the helical backbone (EC stage), via the cyclic skeleton (SC stage) to the toroid main chain (cluster stage). The eccentricity of its ellipsoid model is drastically fluctuated (0.96 ~ 0.11) and is equilibrated at 0.47 ~ 0.57, which is obviously lower than that of *meso*-PDHG-F (shown in Fig. 6b). Both collapsed main chains, distinguished from traditional globule manners[51], were predicted merely with coarse-grained models[48]. During the collapsed process, *meso*-PDHG-F undergoes the rotation of Cz groups toward *syn*-type DHG edges to crumple the chains, whereas the rotation between C$_2$-symmetric *meso*-DHG units and fluorenyl groups maintains the linear backbones. Conversely, the chain collapse of *rac*-PDHG-F originates from the rotation between the folded *rac*-DHG units and fluorenyl groups. The above distinguished conformational transformations result in various covalent energies (valence in Fig. 6c) that are linked to the structural stability of collapsed conformations. For example, due to structural instability stemming from *syn*-type DHG edges, the collapsed *meso*-PDHG-F possesses higher average covalent energy (7.3 kcal mol$^{-1}$ per repeat units) than that of collapsed *rac*-PDHG-F (Fig. 6c). This result also suggests that the *meso*-configured repeat units enhance the ability to prevent chain collapse, which indirectly increased the backbone rigidity vs. *rac*-configuration. Further, these features are majorly maintained in other chain length (DP = 7, 9, and 18, Supplementary Figs. 27–29), ruling out the entropic effect on such chain relaxation[49]. In terms of above collapsed behaviors, PDHG-F chains in full *meso*-configuration are more likely to serve as the rigid nanopolymer models.

## Discussion

We demonstrated the gridization rule of A$_n$B$_m$-type synthons under different internal and external conditions, and discovered the stereoselective gridization that is kinetically controlled rather than thermodynamically controlled processes during ultrafast superelectrophilic Friedel–Crafts reaction. As a result, *meso*-configurational Drawing Hand-type gridarenes are efficiently synthesized and confirmed by various tools such as single-crystalline X-ray diffraction, NMR, FT-IR, and so on. The stereo-determining step stems from the centrosymmetric packing of superelectrophilic species via synergistically π–π stacking attractions and intermolecular multicationic repulsions. We further examine the gridization rule of A$_2$B$_2$-type monomers into polygrids that are an alternative to synthetic methodology of ring-chain-alternating polygridarene backbones. As a result, such *meso*-stereocontrol is genetically transformed into polygridization under the initial stage that support the hypothesis of antiparallel molecular packing modes. To understand tacticity of polygrids, molecular dynamic simulations are made to visualize the differences of chain collapse between *meso*- and *rac*-polygrids. Compared with *rac*-polygrids, *meso*-polygrids exhibit rod-like conformations and increased ability against chain collapse, which reveals higher backbone rigidity. Moreover, these stereoregular nanopolymers exhibit microscopic dynamics distinctive from traditional polymers, which will fundamentally impact on the polymer science. Our gridization and polygridization afford powerful molecular installing nanotechnology for the innovation of covalent nano-linkages with the multi-bonding feature and the multiscale molecules with the potential application of organic electronic materials, devices, and intelligence.

## Methods

**Materials**. The synthetic procedures and details of the Friedel–Crafts reaction, gridization, and polygridizations are described in the Supplementary Information files. The preparation of substrates are detailed in the Supplementary Information files as well.

**Structural characteristics via NMR technologies**. $^1$H and $^{13}$C NMR spectra were acquired from a Bruker 400 MHz NMR Fourier transform spectrometer (400 MHz and 100 MHz, respectively) at 20 °C. The *meso*-DHG4, *rac*-DHG4, and PDHG-F samples were characterized by the $^1$H NMR Fourier transform spectrometer in 600 MHz. Chemical shifts are shown as $\delta$ in units of p.p.m. relative to an internal standard [$^1$H NMR: tetramethylsilane = 0.00 p.p.m.] or relative residual peaks ($^1$H NMR: 7.26 for CDCl$_3$, 2.50 for $d_6$ -DMSO; $^{13}$C NMR: 77.0 triplet for CDCl$_3$, 39.25 for $d_6$-DMSO. In brief, multiplicities of every signal peak are shown as follows: s (singlet); d (doublet); t (triplet); q (quartet); dd (doublet of doublets); dt (doublet of triplets); m (multiplet). Coupling constants are expressed as a $J$-value in Hz. To calculate the *meso*-selectivity during the gridization, the reaction solution was quenched with KOH aqueous solution and then was extracted with CH$_2$Cl$_2$ solvents. Then, we heated the organic phase to remove the CH$_2$Cl$_2$ solvent and added CDCl$_3$ solvent to perform the NMR measurements. $^1$H-$^1$H correlation spectroscopy ($^1$H-$^1$H COSY) was performed to assign the protons in *meso*-DHG4, protonated *meso*-DHG1, and PDHG-F. NOESY was performed to demonstrate the likely stacking mode of protonated MC1 and MC2.

**Measurements of molar mass with mass spectrometry**. Matrix-assisted laser desorption/ionization time-of-flight mass spectrometry was utilized to obtain the molecular mass of substrates. The DHG products with 900 ~ 1100 m/z and small molecules (<750 m/z) were characterized by High Resolution Mass Spectrometry, in brief, HRMS (Thermo Fisher Scientific, Linear ion Trap Fourier Transform ion Cyclotron Resonance Mass Spectrometry, in brief, LTQ-FTICR-MS).

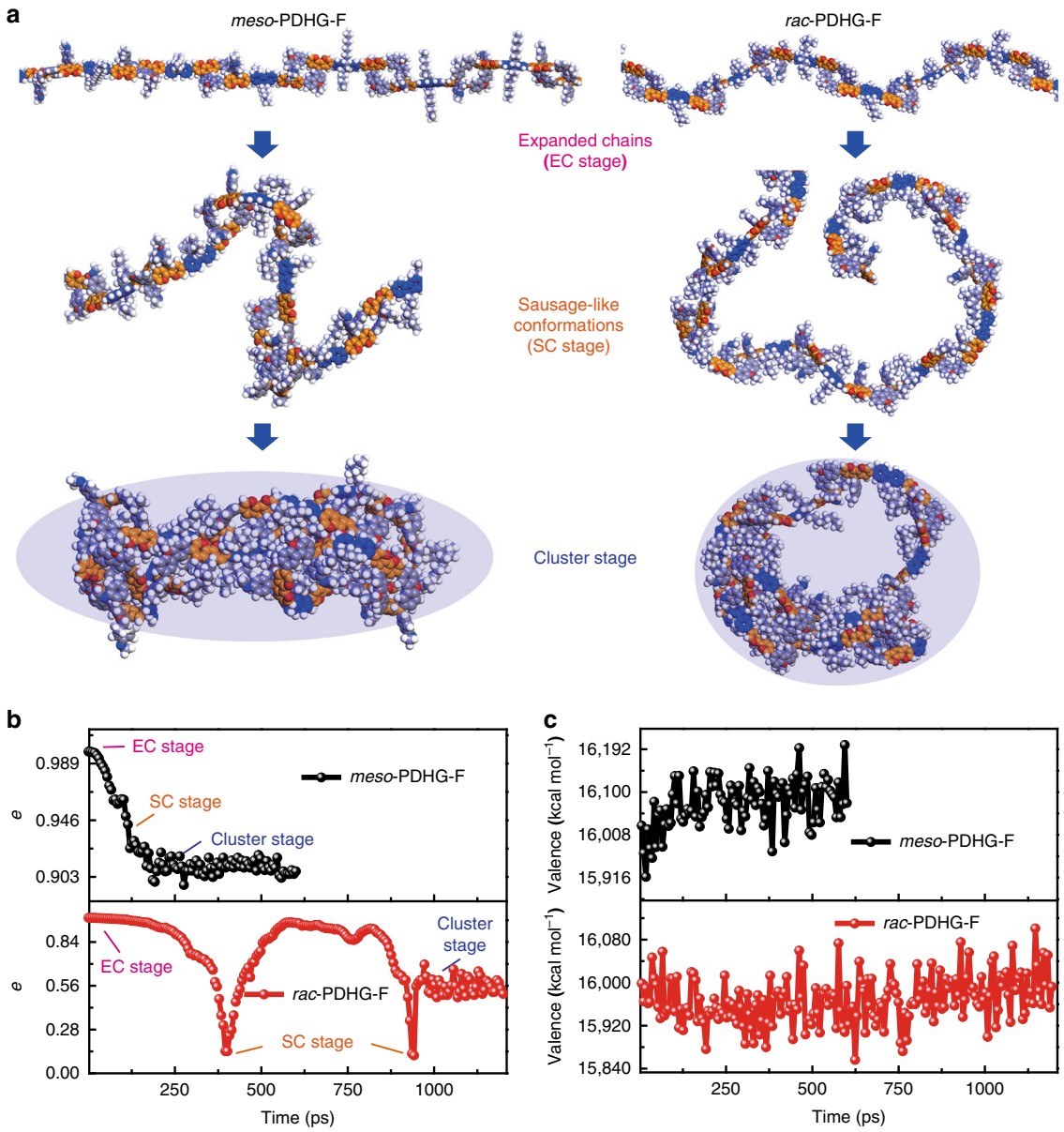

**Fig. 6 The molecular dynamic simulation of the single PDHG-F chain collapse under the vacuum conditions (298 K).** *Meso*-PDHG-F is defined as the main chain with 100% *de* of *meso*-configuration; *rac*-PDHG-F polygrids is represented as the backbone exhibiting 100% *de* of *rac*-configuration. **a** The conformations (DP = 13) of expanded chains (EC stage), sausage-like chains (SC stage), and collapsed chains (CC stage) during the collapsed process. The blue region displays the ellipsoid model of PDHG-F backbones that completely covers the main chain. **b** The eccentricity (*e*) of the ellipsoid model (blue region) for quantitatively evaluating the conformational anisotropy (*e* = 1 for the 100% anisotropy and *e* = 0 for the perfect isotropy) of *meso*-PDHG-F (black) and *rac*-PDHG-F chains (red). **c** The covalent energy (valence) of the main chain during the collapse of the *meso*-PDHG-F (black) and *rac*-PDHG-F (red) backbones.

**Single-crystal crystallography**. X-ray diffraction was performed on a Bruker D8 X-ray diffractometer with Cu Kα radiation ($\lambda = 1.54050$ Å). The operating 2θ angle was in the range of 5 ~ 30° with a step length of 0.02°. The single-crystal crystallography data were all collected at ~100 K on a Bruker 2000 CCD area detector with graphite-monochromated Mo Kα radiation ($\lambda = 0.71073$ Å). To solve the structures of *meso*-DHG1, *meso*-DHG4, *rac*-DHG1, *rac*-DHG2, and **2a**, we utilized direct methods in SHELXS-2015 and refined against F2 via SHELXL-2015. Hydrogen atoms were fixed geometrically and refined isotropically.

**Characteristics via FT-IR method**. The DHGs and PDHG-F products were addressed as KBr pellets. For the mixing systems ($de = 0 \sim 67\%$), each system was obtained by mixing *meso*-DHG1 with *rac*-DHG1 sample in 1:1, 2:1, 3:1, 4:1, and 5:1 molar ratio. Then the infrared absorptions of these samples were measured by a Fourier transform infrared spectrophotometer (SHIMADZU, IRPrestige-21).

**GPC measurements**. GPC experiments were conducted on an HP1100 HPLC system possessing 7911GP-502 and GPC columns using PSs as the standards and tetrahydrofuran (THF) as the eluent at a flow rate of 1.0 ml min$^{-1}$ at 25 °C. The corresponding principles are detailed in the Supplementary Information.

**DLS experiments**. The DLS characteristics were determined by a Brookhaven instrument (BI-200SM). The concentration of PDHG-F samples were lower than 0.05 mg ml$^{-1}$ because of the existence of aggregation that drastically amplifies the results of hydrodynamic radius.

**Visualization via atomic force microscopy**. The atomic force microscopy characteristics were performed under the peak-force tapping mode. In order to reduce the tip-broadening effect, we selected the supersharp tips (type: SAA-HPI-SS). The PDHG-F film on the mica sheet was via spin-coating the solutions (concentration: 3 µg ml$^{-1}$; solvent: 1,2-dichloroethane).

**Single energy of DHG1 conformations**. We use the software Material Studio 2016 to calculate single energies of DHG1 conformations I ~ IV (in Fig. 5a) via the DMol3 calculation module. The theoretical calculation is based on the method of B3LYP. The transition state (TS) method was used for the density functional theory with van der Waals dispersion corrections (DFT-D correction).

**Atomistic simulation of chain collapse**. The software Material Studio 2016 was utilized to simulate the chain collapse. Before performing the dynamics, all PDHG-F chains were constructed in the expanded conformations where the dihedral angle between adjacent repeating units are approximate to 180°. The dynamics of chain collapse were carried out under the conditions of NVT ensemble (298 K) and the COMPASS forcefield (the summation method of van de Waals term and the electrostatic term are both Group-based in which the cutoff distance were set as 12.5 Å). The total times of dynamics were set as 600 ps for *meso*-PDHG-F (DP = 13) with the time step of 1.5 ps. In fact, the *meso*-PDHG-F falls into collapsed equilibrium (cluster stage) after 100–250 ps. By contrast, due to much slow relaxation process toward collapsed equilibrium, the total times of dynamics were set as 3000 ps for *rac*-PDHG-F (DP = 13) with the time step of 1.5 ps.

## Data availability

The authors declare that all data supporting the current findings of this study are available in the main manuscript or in the Supplementary Information. The single-crystal datum (including a.cif file and a structural figure) of **2a**, *meso*-DHG1, *rac*-DHG1, *rac*-DHG2, and *meso*-DHG4 were upload in Cambridge Structural Database, obtaining corresponding CCDC number 1886054, 1886046, 1886053, 1886052, and 1938823, respectively. The source data underlying Figs. 5a and 6a are provided as a Source Data file. Other data are available from the corresponding author on reasonable request.

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

## Acknowledgements

We thank Xueyan Liu from Shiyanjia Lab (www.Shiyanjia.com) for the COSY and [1]H NMR characteristics. We thank Jinwei Liu from Shiyanjia Lab (www.Shiyanjia.com) for the COSY, HRMS, and NOESY characteristics. We thank daxia Xia from Shiyanjia Lab (www.Shiyanjia.com) for the simulation of the vibrational modes. We thank Chenqing from Shiyanjia Lab (www.Shiyanjia.com) for the DLS characteristics. We thank Changjin Ou for single-crystal crystallography analysis. This work was supported by the National Natural Science Foundation of China (21774061, 21602111 and 61935017), National Natural Science Foundation major research program integration project (Grant Number 91833306), the Natural Science Foundation of Jiangsu Province of China (BK20150832), Synergetic Innovation Centre for Organic Electronics and Information Displays, Six Peak Talents Foundation of Jiangsu Province (XCLCXTD-009), and Postgraduate Innovation Fundation of Jiangsu Province (46030CX17747 and SJCX18_0302).

## Author contributions

D.L. has prepared above substrates, performed all of gridizations and polygridizations, analysed all of datum (in terms of stereoselectivity and polymeric characterizations), performed the simulation of chain collapse, and written this manuscript and Supplementary Information. Y.W. has carefully polished up this manuscript and Supplementary Information files, as well as provided crucial insight into the organic synthesis of this work. A.P., He Zhang, C.Z., and X.Z. have performed parts of experiments in terms of the synthesis of substrates. D.L. and Hao Zhang have provided the characterization of PDHG-F samples. L.Y. has provided the theoretical calculation for the geometric optimization of *meso*-DHG1 and *rac*-DHG1. Q.F. has provided essential suggestions for this work. L.X. and W.H., both of whom are corresponding authors, have initiated the project, provided enough funds to ensure this work, and supervised every step of the work. Especially, L.X. has created the Drawing Hands pictures.

## Competing interests

The authors declare no competing interests.
