## [Peer Review File · Nature Communications]

Reviewers' comments:

Reviewer #1 (Remarks to the Author):

This manuscript describes the synthesis of molecular grids, and in particular, a polymeric derivative of the molecular grid. The work is an extension of previous papers from this research group (i.e., references 23-26). Synthetic studies are initiated with a diazafluorene and its Friedel-Crafts reaction with a carbazole substrate (Fig. 2a). The authors obtain a mixture of single and double substitution products at the carbazole. The double substitution products (3) are formed as the major product in up to 85% yield. Although these products should be formed as a mixture of diastereomers (meso/DL), the authors do not comment on the formed stereoisomeric ratios. At this point, the authors studied the effects of some substituents and acid-promoters on the success of the reactions.

Following these initial studies, the authors prepared a substrate for gridization, that is a substrate having both an electrophilic and nucleophilic site (MC). When this substrate is reacted in acid, the dimeric products are formed (DHG) – the basic unit of the molecular grid. This dimerization chemistry is similar to that previously described in reference 24, although different aromatic nucleophiles and electrophiles are used. Both dimeric products are well characterized spectroscopically and the authors were even able to obtain x-ray crystal structures of the two DGH1 dimer diastereomers (meso, DL). In addition to the DHG dimers, the authors isolated varying amounts of cyclic oligomers (MG). Substituent effects were also examined for these conversions.

The authors then attempted to carry out the synthesis of polygridization macromolecules using a series of substrates (DC) which could form the molecular grid and form a polymeric structure during this chemistry. Essentially, this is an A2B2 polymerization. Reaction of the DC substrate with acid does appear to provide macromolecules with the grid substructure but it also can produce cross-linked polymeric structures (CP) from random coupling reactions. Based on NMR analysis, the authors claim that the chemistry leads to stereoregular polymers. For example, the substrate DC-F is claimed to possess meso-DGHs in 95% proportion. The rationale for this assignment is explained on page 9, paragraph 1 of the manuscript.

This work was followed by further characterization of the macromolecules, including GPC characterization to estimate its degree of polymerization and end group analysis. Further, the rigidity of the chains were estimated using light scattering techniques.

Finally, the authors discuss mechanisms of chain growth. This is coupled with conformational studies of the monomeric units and energy calculations of the cyclization intermediates. They propose a model for the high meso-gridization.

In general, this is nice chemistry derived from an innovative idea. Unfortunately, it is the opinion of this reviewer that the study does not meet the demands for publication in Nature Communications. Here is why:

1. In my inspection of the raw data, there were some problems. In previous papers from this group, the NMR spectra of synthesized compounds were almost always perfect. Not so in this study. For example with compound 3a, the ¹H NMR analytical data describes the ethyl group as having a triplet, 3H, 1.22 ppm, and a multiplet, 2H, 4.43 ppm. Mysteriously, the integration of the 2H peak is not included on the spectrum. If it had been included, it would be obvious that it could not be the methylene group. Similarly, the ¹³C NMR is really poor quality. Although the authors try to assign a peak at 52 ppm, it is hard to see why this is not simply a noise peak. In other spectra, there are impurities that should be easily removed by vacuum (such as DCM) but they were not removed. Likewise, a number of the spectra had unexplainable "extra" peaks. For example, meso-DHG1 is described in the ¹³C NMR as having five peaks in the aliphatic region

(12.9, 30.2, 37.9, 55.1, 63.3). Yet, the C₂ symmetry of this compound should only lead to four such peaks.

2. A large part of this manuscript is based on the claim that the polygridization may occur in up to 95% selectivity for the meso. This is based on the integration of a single peak in the NMR of PDHG-F (Figure 3). Specifically, the authors are examining proton i around 8.3 ppm. Since this polymer only appears to have the i proton at 8.3 ppm (similar to the meso-DHG4) and not an i proton absorbance at 8.5 ppm (similar to the rac-DHG4), then this is assumed to be evidence for stereoselective polygridization. There are two serious problems with this. First, subtle changes in the grid conformation could lead to flattening of the i peak arising from the rac-form of PDHG-F. Secondly, the authors seem to ignore the contradictory evidence that arises from the 4-methoxyphenyl protons (protons a and b). Thus, the meso-DHG4 has a and b at 6.7 and 6.9 ppm, while the rac-DHG4 has a and b nicely separated at 6.8 and 7.2 ppm. Examination of the PDHG-F NMR shows BOTH SETS OF PEAKS. Although they do appear to be in roughly equal amounts, it is the rac-PDHG-F that is the (slightly) favored stereoisomeric form. Likewise, the broad peak from N6/N3 would also be best explained from overlap of the corresponding N6/N3 peaks of the DHG isomers. These considerations suggest that both the meso and rac stereoisomers are formed upon polygridization – contrary to the authors' claims.

3. While this is a good idea and the chemistry could have immense practical value, it may also be considered a logical next step from work that is already published. I really enjoyed reading this paper, but nevertheless, it lacks the high level of novelty that is required for a journal such as Nature Communications.

Other considerations for the authors:

1. In reactions done with excess acid, there are few if any hydrogen bonding sites available for stereoselective control as is suggested in the manuscript (due to protosolvation). A more likely element of stereocontrol is the pi-stacking between electron rich aryl groups and the cationic aryl groups.

2. In the reactions done with highly acidic conditions, the Friedel-Crafts reactions are possibly reversible. This may provide a basis for the thermodynamic control of product distribution. With less acidic conditions, the reversibility of the Friedel-Crafts reactions becomes unlikely.

Reviewer #2 (Remarks to the Author):

The manuscript describes a quite interesting system of cyclisation/polymerisation using some unusual monomers. The work is well performed with the products characterised correctly. The manuscript describes the work reasonable well, although the point of the work is unclear and the gridarene nomenclature is ill-defined and appears unnecessary. The manuscript would benefit from telling the reader why this work was carried out and what will be done with the results now they have been obtained.

These should be added to any revised manuscript

Note for answer the questions from reviewers

Reviewer 1:

This manuscript describes the synthesis of **molecular grids**, and in particular, a polymeric derivative of the molecular grid. The work is an extension of previous papers from this research group (i.e., references **23-26**). Synthetic studies are initiated with a diazafluorenol and its Friedel-Crafts reaction with a carbazole substrate (Fig. 2a). The authors obtain a **mixture** of single and double substitution products at the carbazole. The double substitution products (3) are formed as the major product in up to 85% yield. Although these product should be formed as a mixture of diastereomers (meso/DL), the authors do not comment on the **formed stereoisomeric ratios**. At this point, the authors studied the effects of some substituents and acid-promoters on the success of the reactions.

Following these initial studies, the authors prepared a substrate for gridization, that is a substrate have both an electrophilic and nucleophilic site (MC). When this substrate is reacted in acid, the dimeric products are formed (DHG) – the basic unit of the molecular grid. This dimerization chemistry is similar to that previously described in reference 24, although different aromatic nucleophiles and electrophiles are used. Both dimeric products are well characterized spectroscopically and the authors were even able to obtain **x-ray crystal structures** of the two DGH1 dimer diastereomers (meso, DL). In addition to the DHG dimers, the authors isolated varying amounts of **cyclic oligomers (MG)**. Substituent effects were also examined for these conversions.

The authors then attempted to carry out the synthesis of polygridization macromolecules using a series of substrates (DC) which could form the molecular grid and form a polymeric structure during this chemistry. Essentially, this is an **A2B2 polymerization**. Reaction of the DC substrate with acid does appear to provide **macromolecules** with the grid substructure but it also can produce **cross-linked polymeric structures (CP)** from random coupling reactions. Based on NMR analysis, the authors claim that the chemistry lead to **stereoregular polymers**. For example, the substrate DC-F is claimed to possess **meso-DGHs** in 95% proportion. The rational for this assignment is explained on page 9, paragraph 1 of the manuscript. This work was followed by further characterization of the macromolecules, including GPC characterization to estimate its degree of polymerization and end group analysis. Further, the rigidity of the chains were estimated using light scattering techniques.

Finally, the authors discuss mechanisms of chain growth. This is coupled with **conformational studies** of the monomeric units and **energy calculations** of the **cyclization intermediates**. They propose a model for the high meso-gridization.

In general, this is **nice chemistry derived from an innovative idea**. Unfortunately, it is the opinion of this reviewer that the study does not meet the demands for publication in Nature Communications.

Thank you for your careful comments and suggestions. Gridarenes are grid-shaped closed building blocks with extendable, installable, scalable and portable features. Although this work is seemingly similar to the previous work in reference 23 (*Mater. Chem. Front.* **1**, 455-459 (2017)) and 24 (*Eur. J. Org. Chem.* 2018, 48, 7009-7016.) in terms of the A₁B₁-typed substrates. However, I have to stress that the gridization pathways are much

sensitive to the substitutes (on the alcohol-based substrates) including their configuration shapes, the bend angle and the length between the alcohol and the benzoid reaction sites (even under the same A_nB_n models). For example, if the benzoid group is replaced by thiophene segments on the A_1B_1 -type I-shape substrate, no dimerized grid products will be generated while massive trimerized (three stereoisomers with two diastereoisomers), tetramerized (six stereoisomers with four diastereoisomers) and other oligomerized windmill-like grids will be obtained. However, if this substrate is improved to be longer and is transformed to be L-shape (still in A_1B_1 -type), the dimerized gridization will be improved to ~50% (referring to the literature 24, as the tetramerized windmill-like gridarenes with four diastereoisomers). **However, the stereoselectivity were still demonstrated to be poor in previous work**, which afforded a series of chiral isomers that are isolated in drastic difficulty. When the carbazole moiety is incorporated in the A_1B_1 -type substrates, there are three linking modes (at 2-sites or 3-sites or 9-sites of the carbazole moiety), which are worth deeply researching in the future. Meanwhile, compared with other A_1B_1 -typed (even with alkoxy groups), the moon-crescent substrates (the carbazole group linked at 3-site) **MC** enables to use the advantage of geometric matching that facilitates the formation of dimerized gridarenes with only two chiral centers (two stereoisomers divided into thermodynamic and kinetic gridarene products). In this case, the stereocontrol process can be easier to research and control. Even so, during the Friedel-Crafts process of the dimerized gridization, the powerful thermodynamic equilibrium still leads to the random oligomers and the gridarene products all with the most stable structures. By contrast, some of specific gridarene stereoisomers (such as meso-configuration herein) from kinetic reaction pathways are very difficult to efficiently synthesize, although they may have centrosymmetric backbones potential to construct stereoregular pi-nanoarchitectures or nanopolymers with interesting properties. Toward this, we tentatively endowed the moon-crescent substrates with powerful supramolecular interactions (synergistic by hydrogen-bonding attraction and cationic repulsion) to form the specific intermolecular packing mode in the reaction solution, which is able to **overcome the thermodynamic gridization limit to efficiently afford the kinetic gridarene products (with less stable structures)**. **The molecular packing of substrates is crucially related to** the diazafluorene (DAF) segments and the methoxy groups, which is detailed in this work. In all, these substrates, with ingenious noncovalent interactions to significantly support the kinetic gridization stereoselectivity, are drastically different from previous ones reported in reference 24.

We take care of the stereoselectivity in the Friedel-Crafts reactions. Because the diazafluorene moieties are symmetric and no chiral centers on them, the products of 2a, 2b, 3a and 3b are achiral. In this case, no stereocontrol process are considered.

1. In my **inspection** of the **raw data**, there were some problems. In previous papers from this group, the NMR spectra of synthesized compounds were almost always perfect. Not so in this study. For example with compound **3a**, the 1H NMR analytical data describes the ethyl group as having a triplet, 3H, 1.22 ppm, and a multiplet, 2H, 4.43 ppm. Mysteriously, the integration of the 2H peak is not included on the spectrum. If it had been included, it would be obvious that it could not be the methylene group. Similarly, the ^{13}C NMR is really

poor quality. Although the authors try to assign a peak at 52 ppm, it is hard to see why this is not simply a noise peak. In other spectra, there are impurities that should be easily removed by vacuum (such as DCM) but they were not removed. Likewise, a number of the spectra had unexplainable “extra” peaks. For example, *meso*-DHG1 is described in the ^{13}C NMR as having five peaks in the aliphatic region (12.9, 30.2, 37.9, 55.1, 63.3). Yet, the C2 symmetry of this compound should only lead to four such peaks.

Thank you for your comments on the NMR results. The ^1H NMR spectra of compounds that contain DCM solvents have been remeasured such as compounds **MC3**, *meso*-DHG1, *rac*-DHG1, *meso*-DHG3 and *meso*-DHG4. However, some of the solvents are much difficult to remove. For example, the triethylamine (used in the eluent solvents of column chromatography) almost cannot be removed from **MG** and their derivatives (extremely large polarity in the column chromatography). We hypothesized the triethylamine can complex with the **MG** derivatives firmly. Some of compounds have poor solubility such as **MC1**, **MC2**, **MC3**, *meso*-DHG1, *meso*-DHG2 and *meso*-DHG3 but they can be soluble in high concentration if adding CF_3COOH (1/3 volume of the CDCl_3 solvent). As a result, we can obtain their ^{13}C NMR spectra of their protonated form. For the ^{13}C NMR spectra, the *meso*-DHG1 does exhibit four signals in the ranges of 70~5 ppm (shown in **Fig. R1**), including 63.6, 55.4, 38.1 and 13.0 ppm that are assigned to the carbon at the 9-site of DAF groups, methoxyl groups, the methylene segments and the methyl groups on the carbazoles respectively.

Fig. R1. The ^{13}C NMR spectra of *meso*-DHG1

Fig. R2. The emulsion of 3a-DMSO system

Unfortunately, the compound **3a** and **3b** exhibit extremely poor solubility even if adding CF_3COOH . For compound **3a**, only a little materials are soluble in DMSO but the emulsion is also exhibited (**Fig. R2**).

Fortunately, we obtain the relatively unambiguous ^1H NMR spectra in the d_6 -DMSO solvent (though the insoluble sediment existed during the characterization).

Fig. R3. The ^1H NMR spectra of 3a. The proton assignment is provided as well.

The hydrogen signals of methylene on the carbazole can be only assigned to be at 4.29 ppm whose chemical shift is consistent with the 2a and 3a compound. However, the drastically large integration indicates that this signal should not be assigned to the methylene (inconsistent with the molecular structure). Furthermore, such signal is a singlet peak, which can be assigned to the solvent impurity. **The methylene signal can be overlapped with this solvent peak. Nevertheless, we can still assign other peaks that testify this structure** (**Fig. R3**). For example, the 8.85 ppm peak with 5.2 Hz of coupling constant is assigned to N3/N6 positions of DAF groups. The 8.46~8.44 ppm peak ($J = 8.0$ Hz) is assigned to N1/N8-sites on the DAF groups. The single signal at 7.88 ppm is merely assigned to be d-sites on the carbazole group because no other hydrogen atoms belong to the singlet signal. The dd peak of the signal at the 7.78 ~ 7.75 ppm is assigned to N2/N7-sites on the DAF groups because their two coupling constant 5.2 Hz and 8.0 Hz are corresponding to the coupling of N3-N2 (or N6-N7) and N2-N1 (or N7-N8). The doublet peak at 7.62 ~ 7.59 ppm is assigned to f or g-sites at the carbazole group. The triplet signal

at the 1.23 ~ 1.20 ppm is assigned to the methyl group on the carbazole segment. What's more, the integration ratio of N3/N6 to i (meanwhile, N3/N6 to f or g) is equal to 2. Similarly, the integration ratio of N3/N6 to methyl group is equal to 4:3 instead of 2:3. Therefore, the 3a structure is the disubstitution of DAF on the ethyl-carbazole in the Friedel-Crafts reaction.

Fig. R4. The ^{13}C NMR spectra of **3a**. The sp^3 -hybridized carbon atoms are assigned.

For the spectra of **3a**, the peak at the 52 ppm is the noise signal indeed. **The powerful peak at the 40 ppm should be assigned to DMSO (39.52 ppm) and the methylene signal is likely to overlapping with the DMSO (Considering that such carbon signal is at 37.6 ppm for 2a in CDCl_3 solvent).** In the ranges of 0-80 ppm, the signals at the 62.8 and 14.1 are assigned to the quaternary carbon atom (at the DAF groups) and the methyl group (at the carbazole group). These chemical shift of these signals is consistent with **2a**. In all, this **3a** compound confirms the disubstitution of DAF groups during the Friedel-Crafts reactions.

This explanation has been supplemented in the supplementary information.

2. A large part of this manuscripts is based on the claim that the polygridization may occur in up to 95% selectivity for the meso. This is based on the integration of a single peak in the NMR of PDHG-F (Figure 3). Specifically, the authors are examining proton i around 8.3 ppm. Since this polymer only appears to have the i proton at 8.3 ppm (similar to the meso-DHG4) and not an i proton absorbance at 8.5 ppm (similar to the rac-DHG4), then this is assumed to be evidence for stereoselective polygridization. There are two serious problems with this. First, subtle changes in the grid conformation could lead to flattening of the i peak arising from the rac-form of PDHG-F. Secondly, the authors seem to ignore the contradictory evidence that arises from the 4-methoxyphenyl protons (protons a and b).

Thus, the meso-DHG4 has a and b at 6.7 and 6.9 ppm, while the rac-DHG4 has a and b nicely separated at 6.8 and 7.2 ppm. Examination of the PDHG-F NMR shows BOTH SETS OF PEAKS. Although they do appear to be in roughly equal amounts, it is the rac-PDHG-F that is the (slightly) favored stereoisomeric form. Likewise, the broad peak from N6/N3 would also be best explained from overlap of the corresponding N6/N3 peaks of the DHG isomers. These considerations suggest that both the meso and rac stereoisomers are formed upon polygridization – **contrary to the authors claims**.

Thank you for your very careful suggestions on the discussion about the stereoselectivity.

Fig. R5. The ¹H NMR spectra of meso-DHG4, rac-DHG4, MG1 and the PDHG-F (2.5 equiv of MC1)

First of all, I want to detail the origin of the special signals at 6.80 ppm (in fact, the chemical shift should be 6.75 ppm) and 7.20 ppm. **We found that these peaks should be assigned to MG1 that arises from the excessive end-capping reagent MC1.** In the polygridization, we added the massive MC1 (2.5 equivalents to the DC-F monomer) to realize the instantaneous termination in which the MG1 can be generated from MC1 that do not terminate with the polymer intermediates. Shown in Fig. R5, **most of additional signals are more consistent with MG1 rather than meso-DHG4.** To confirm this point, we performed the polygridization of DC-F with less MC1 end-capping reagent, 0.5 equiv to DC-F monomers (or even 0.25 equiv to DC-F). Fortunately, this polygridization still underwent so smoothly that no insoluble cross-linking polymers CP-F were generated.

Fig. R6. The ^1H NMR spectra of PDHG-F with different batches (obtained from the polygridization with 2.5, 0.50, 0.55 and 0.25 equiv of MC1 reagents. The samples that are terminated by the 0.5 and 0.25 equiv of MC1 respectively have been characterized by 600 MHz ^1H NMR).

For the PDHG-F products from these reactions, we analyzed their ^1H NMR spectra in Fig. R6. Luckily, we found that **lowering the MC1 amount did diminish and even eliminate the hydrogen signals at the 7.20 and 6.80~6.75 ppm that was misunderstood as the rac-configuration.** In addition, ranging from 9.20 to 8.90 ppm, the dominant peaks are located at the 9.12 and 8.97 ppm that are consistent with the meso-selectivity. The broad peak at the 8.70-8.65 ppm (c-site on the carbazole group) in the samples terminated by 2.5 equiv of MC1 is obviously eliminated (which can be mistaken as the rac-configuration as well). It is noted that **the integral ratio of the peak at the 8.30 ppm to at the 8.62 ppm is about 2:1 for the samples terminated by 0.25~0.5 equiv of MC1, which is absolutely in agreement with meso-DHG4 (8.30 ppm is assigned to be i and N8-positions while 8.62 is assigned to be c-sites).** On the contrary, the PDHG-F (using 2.5 equiv of MC1) is about 1:1, inconsistent with the meso-DHG4 probably because of the MG1 residue. **As a result of these eliminated peaks, such PDHG-F (terminated by 0.25~0.5 equiv of MC1) displays the dominant meso-form.**

Fig. R7. The ^1H NMR spectra of PDHG-F with different batches (The UT-PDHG-F sample is from the polygridization that did not terminate with MC1 and quenched directly. The SP-PDHG-F is from the Suzuki coupling polymerization via meso-DHG3).

In order to testify such differences that arise from the amount of MC1 reagents (termination), we studied the ^1H NMR spectra of UT-PDHG-F originating from the polygridization without termination (Fig. R7). Fortunately, the peaks at the 7.20 and 6.80~6.75 ppm almost disappear, which is consistent with the PDHG-F (using 0.5, 0.55 and 0.25 equiv of MC1 to terminate the polymerization). What's more, the peak features of N3, N6, c, i and N8 (including the integration ratio) are similar to PDHG-F (0.5 equiv of MC1, 600 MHz) rather than the 2.5 equiv of MC1 termination. Thus, the artifacts of the PDHG-F (using 2.5 equiv of MC1) is based on the massive MG1 residue. In addition, according to the UT-PDHG-F spectra with dominant meso-configuration features instead of rac-form, these polygridization does exhibit meso-selectivity.

To confirm the meso-selectivity further, we tried to synthesize the SP-PDHG-F via the suzuki coupling polymerization of meso-DHG3. For its spectrum (in Fig. R7), the signals at 6.80-6.75 ppm and 7.20 ppm are almost eliminated. Although with slightly up-shift, the peak features of a, b and c-sites are similar to the PDHG-F (using 0.25~0.5 equiv of MC1). Moreover, the integration ratio of the peak at the 8.28 ppm to at the 8.60 ppm is about 2 that is the same as the PDHG-F (using 0.25~0.5 equiv). Because of these evident features, such PDHG-F (using 0.25 equiv of MC1) does exhibit dominant meso-configuration and the polygridization displays the meso-selectivity. The proper ^1H and ^{13}C NMR of PDHG-F (using 0.25 equiv of MC1) are added in the supplementary information to replace the mistaken spectra.

The review also discussed that the “subtle changes in the grid conformation could lead to flattening of the i peak”. In order to testify whether these signals are flattened, we studied all the rac-DHGs spectra.

Fig. R8. The ^1H NMR spectra of *rac*-DHG series (aromatic region). The *rac*-DHG4 has been further purified via HPLC.

As is shown in Fig. R8, the signals of *i* and N8, are very sharp for *rac*-DHG3 and *rac*-DHG4 (repeating units). Even for *rac*-DHG1 with hindered rotation, the *i* and N8 signals are still sharp. Thus, even with subtle conformational changes, the *i* and N8 signals are much more likely to maintain the sharp peak. On this basis, if *rac*-DHG repeating units had existed on the PDHG-F main-chain, these sharp signals ranging from 8.55 to 8.40 ppm would be exhibited as well.

Moreover, in order to rule out the probability that *i* and N8 signals on the *rac*-configuration upshift to 8.30 ppm (overlapped with *i* and N8 signals of *meso*-selectivity), we researched the chain conformation of the PDHG-F (using 0.25~0.5 equiv of MC1). Commonly, the upshifting signals on the chain backbones arise from the intramolecular stacking (like polycatenanes, see the literature Science 358, 1434-1439 (2017)) that comes from the chain collapse. In this term, if the chain collapse had underwent, other hydrogen signals (including DAF, methoxybenzene and *c*-sites on the carbazole groups) would upshift as well. However, these signals maintain the same chemical shift as the repeating units *meso*-DHG4, indicating that the chain collapse does not occur. Thus, *i* and N8 signals should not upshift and overlap with the *meso*-form signals.

In addition, we studied the chain conformation of PDHG-F via the molecular dynamic simulation. Based on the Mark-Houwink exponent $\alpha = 1.651$ in the tetrahydrofuran solvent, such polymer chains exhibit the rigid rod-like conformation (expanded chains) rather than the collapsed chains with very low α (should lower than 0.5). For the relatively good solvent CHCl_3 , they can interacted with the PDHG-F chains and amplify the excluded volume effect to expand the chain backbones. To demonstrate such point, we referred to the equation ES1 (J. Chem. Phys. 1949, 17, 303): $C(1/x_s - 2\chi)N^{1/2} = \beta^5 - \beta^3$, where C is the length ratio of the repeating units to Kuhn segments (defined as the equivalent chains that consist of rigid segments via the linkage of flexible covalent bonds) and N is the number of Kuhn length. The x_s is defined as the volume ratio of the solvent molecules to the repeating units. As the volume of CHCl_3 (V_C) is calculated to be 70.30 \AA^3 ; the volume of tetrahydrofuran (V_T) is

computed to be 79.55 \AA^3 ; the volume of the repeating units of PDHG-F is evaluated to be 1261.86 \AA^3 , $x_s = 0.056$ for CHCl_3 solvent and 0.063 for the tetrahydrofuran solvent are accomplished respectively. The β is defined as the expansion factor in which the $\beta > 1$ corresponds to the expanded chains while $0 < \beta < 1$ belongs to the chain collapse. Commonly, the β ranges from 0.33 to 0.7 if the chains are collapsed (Macromolecules 1992, 25, 1618-1620; Macromolecules 1995, 28, 1049-1059). According to the equation **ES1**, the β value is determined by the Flory-Huggins interaction parameter χ where the high χ indicates the poor compatibility between the solvent and the polymer chains while the low χ suggests such good compatibility. In this case, the high χ leads to dominant intramolecular interactions between polymeric segments, which causes the chain collapses further. On the contrary, the low χ gives rise to dominant interactions between the solvent molecules and the repeating units, which enhances the excluded volume effect and supports the chain expansion or stretching. The χ can be calculated from the equation **ES2** (according to the references: Fluid Phase Equilib, 2002, 203, 247–260): $\chi = V(\delta_P - \delta_S)^2/kT$, where V is the monomeric molar volume (1261.86 \AA^3 for **PDHG-F**); k is the Boltzmann constant; T is the temperature (298 K). As a result, the χ is influenced by the term $(\delta_P - \delta_S)^2$ in which δ_P and δ_S are the solubility parameters of the polymer chains and the solvent molecules respectively. During the molecular dynamic calculation, the solubility parameters are divided into two-dimensional parameters in terms of van de Waals force ($\delta_{P\text{-vdW}}$ and $\delta_{S\text{-vdW}}$) and electrostatic force ($\delta_{P\text{-E}}$ and $\delta_{S\text{-E}}$) according to the calculation method of the COMPASS forcefield (J. Phys. Chem. B 1998, 102, 7338-7364). In this case, **$(\delta_P - \delta_S)^2$ should be transformed to $(\delta_{P\text{-vdW}} - \delta_{S\text{-vdW}})^2 + 0.25 (\delta_{P\text{-E}} - \delta_{S\text{-E}})^2$ that is analogous to the reference** (J. Appl. Polym. Sci., 1993, 50, 513-530). According to the equation $\delta = (\text{CED})^{1/2}$ where CED is the cohesive energy density (involving $\delta_{\text{vdW}} = (\text{CED}_{\text{vdW}})^{1/2}$, $\delta_E = (\text{CED}_E)^{1/2}$, referring to the literature: Polymer, 2010, 51, 291–299), we calculated the CED of PDHG-F chains, the CHCl_3 solvent and the tetrahydrofuran solvent (As is shown in Table **S1**).

Compound	CED_{vdw} (J/m^3)	CED_E (J/m^3)	δ_{vdw} (cal/cm^3) ^{1/2}	δ_E (cal/cm^3) ^{1/2}	$(\delta_P - \delta_S)^2$ (cal/cm^3)
PDHG-F	1.987×10^8	1.016×10^7	6.89	1.56	
CHCl_3	3.204×10^8	6.517×10^6	8.75	1.23	3.49
tetrahydrofuran	3.357×10^8	2.690×10^7	8.95	2.53	4.48

Table S1 the solubility parameters of the **PDHG-F**, CHCl_3 and the tetrahydrofuran. It is noted that **1 cal = 4.187 J**. All CED and δ are the average values in which the standard errors are lower than $0.015 (\text{cal}/\text{cm}^3)^{1/2}$.

In the light of Table **S1**, the term $(\delta_P - \delta_S)^2$ of the PDHG-F and CHCl_3 pair (**3.49**) is relatively lower than the PDHG-F and tetrahydrofuran pair (**4.48**). According to the equations **ES1**, the term $(1/x_s - 2\chi)$ is **8.99** for CHCl_3 solution and **4.35** for the tetrahydrofuran. Considering $C > 0$ and $N > 0$, **the $\beta > 1$ is absolutely satisfied in this case because of the term $\beta^5 - \beta^3 = \beta^3 (\beta + 1)(\beta - 1) > 0$. Moreover, if we further consider $C \approx 1/9$ and $N \approx 1$ for the rigid rod-like conformation of PDHG-F (DP = 9), we will evaluate that $\beta \approx 1.242$ for CHCl_3 solution and $\beta \approx 1.152$ for tetrahydrofuran solution.** Therefore, **the PDHG-F chains are expanded both in CHCl_3 (to more degree) or tetrahydrofuran solutions, while the chain collapse that causes the upshift of i/N8 signals is impossible.** In this case, the chemical shifts of major hydrogen atoms on the

PDHG-F chains should be the same as (or much approximate to) the *meso*-DHG4 or *rac*-DHG4 (DHG4 as the repeating units).

Fig. R9. The ^1H NMR spectra of *meso*-DHG4, PDHG-F and *rac*-DHG4 (in the range of 8.75~8.20 ppm). The *meso*-DHG4 and *rac*-DHG4 serve as the repeating units of the PDHG-F.

As is shown in Fig. R9, we can also observed the very weak signals at the 8.52 and the 8.47 ppm that are consistent with the *i* and N8-sites of the *rac*-configuration respectively. Meanwhile, the weak and broad peak ranging from 8.70 to 8.66 ppm is consistent with the *c*-sites of *rac*-configuration. As the dominant peak of *c*-sites is located at the 8.63 ppm approximate to *meso*-DHG4, along with dominant *i*/N8 signals at the 8.30 ppm, such PDHG-F exhibits the major *meso*-selectivity ($80.1\% \text{ de} = (3.08/2 - 0.17) / (3.08/2 + 0.17)$). If using *c* signals, the *de* of *meso*-selectivity is equal to $(1.48 - 0.17) / (1.48 + 0.17) = 79.4\%$. In conclusion, the ^1H NMR spectra of the PDHG-F (recent prepared samples) powerfully supports the *meso*-selectivity. Therefore, the average *meso*-selectivity is up to about 80% *de*.

Apart from ^1H NMR spectra, the FT-IR spectra confirm the dominant *meso*-selectivity of PDHG-F.

Fig. R10. The FT-IR spectra of the *meso*-DHG1, *meso*-DHG4, *rac*-DHG1 and *rac*-DHG4 in the range of 1500~600 cm^{-1}

As is shown in Fig. R10, the differences between *meso*-configuration and *rac*-configuration are distinguished. Both ***meso*-DHG1** and ***meso*-DHG4** exhibit the strong vibration absorption at the 1248 (the strongest) and 1294 cm^{-1} while the bands at the 1024 and 1100~1085 cm^{-1} (almost in nonexistence) are weak. On the contrary, the bands at the 1024 and 1100~1085 cm^{-1} become much strong for both ***rac*-DHG1** and ***rac*-DHG4**, which is relatively stronger than that at 1264 cm^{-1} . Furthermore, the 1294 cm^{-1} peak are so weak that they can be hardly observed. In all, the intensity distribution varies evidently between *meso*- and *rac*-configurations.

Fig. R11. The vibration mode of *meso*-DHG1 and *rac*-DHG1, as obtained from the theoretical calculation based on the method of RB3LYP and the basic set of 6-31G(D). The scaling factor is referred to the literature (J. Phys. Chem. 1996, 100, 16502).

To further testify above vibration differences, we simulated the vibration mode of above wavenumbers for *meso*-DHG1 and *rac*-DHG1 respectively (As is shown in Fig. R11). For the band at the 1024 cm^{-1} (assigned to the C-O stretching vibration on the methoxyl groups), such vibration mode possesses the centrosymmetric vibration distribution on *meso*-DHG1 with C_2 -symmetric backbones, which leads to much weaker dipole changes than that of asymmetric *rac*-DHG1 structures. As a result, the infrared activity of 1024 cm^{-1} is stronger for *rac*-DHG1 than that for *meso*-DHG1, giving rise to stronger vibration band in *rac*-DHG1. For the band of $1100\sim 1085\text{ cm}^{-1}$ (C-H scissoring vibrations on the methoxybenzene segments), the strictly centrosymmetric vibration mode leads to extremely low infrared activity on *meso*-DHG1 while the same vibration directions on the asymmetric *rac*-DHG1 afford high infrared activity. Hence, such band exhibits much stronger intensity in *rac*-DHGs. On the other hand, the asymmetric vibration mode of C-O bond (on the methoxybenzene groups, 1248 cm^{-1}) results in high infrared activity in *meso*-DHG1. What's more, the asymmetric C-H scissoring vibrations (1294 cm^{-1}) on the 3,6-sites of DAF groups afford obviously higher infrared activity in *meso*-DHG backbones than that in *rac*-DHG blocks with

centresymmetric vibration distribution (much weaker dipole changes). Thus, the 1294 cm^{-1} band is strong in *meso*-DHG1 but rather weak in *rac*-DHG1.

Fig. R12. The FT-IR spectra of the doping systems where *meso*-DHG1 and *rac*-DHG1 are mixed in specific ratio (0% de = 1:1 of *meso*: *rac*; 33% de = 2:1; 50% de = 3:1; 60% de = 4:1; 67% = 5:1). a) displays the FT-IR spectra of the mixed systems and *rac*-DHG1, *meso*-DHG1 as well as the PDHG-F. b) shows the band intensity ratio of 1248~1264 cm^{-1} to 1024 cm^{-1} (along the y-axis) and the other band intensity ratio of 1294 to 1024 cm^{-1} (along the x-axis).

As is shown in Fig. R12, increasing the amount of *meso*-DHG1 in the pure *rac*-DHG1 systems decreases the band intensity at the 1100~1080 and 1024 cm^{-1} but enhances the vibration at 1294 cm^{-1} . For example, the intensity ratio of 1248~1264 cm^{-1} to 1024 cm^{-1} (I_{1248}/I_{1024}) is increased from 0.558 (*rac*-DHG1), to 0.741 (0% de), 0.954 (33% de), 1.053 (50% de), 1.103 (60% de), 1.196 (67% de) and 1.546 (*meso*-DHG1). Meanwhile, the intensity ratio of 1294 cm^{-1} to 1024 cm^{-1} (I_{1294}/I_{1024}) is increased from 0.127 (*rac*-DHG1), to 0.230 (0% de), 0.340 (33% de), 0.399 (50% de), 0.462 (60% de), 0.554 (67% de) and 0.762 (*meso*-DHG1). Based on these rules, the PDHG-F sample exhibits $I_{1248}/I_{1024} = 1.402$ and $I_{1294}/I_{1024} = 0.662$, which is approximately in the middle between 67% de and *meso*-DHG1. Therefore, the FT-IR analysis confirms about 80% de of *meso*-selectivity for PDHG-F.

Fig. R13. The ^1H - ^1H COSY of PDHG-F (0.25~0.5 equiv of MC1 termination)

The *meso*-stereoregular PDHG-F is reconfirmed by the ^1H - ^1H COSY. **The cross peak (6.90, 6.70) is assigned to the split coupling correlation of a and b-sites that is consistent with *meso*-DHG1 rather than *rac*-DHG4. In addition, the weaker cross peak (6.85, 6.65) is assigned to the correlation of a' and b'-sites on the terminal DHG units, which is in agreement with *meso*-DHG1 as well. By contrast, we do not observe the cross peak (maybe extremely weak) of (7.20~7.15, 6.80~6.75) that belonging to the correlation of a and b-sites on *rac*-DHG4. Therefore, this result supports the dominant *meso*-configuration of PDHG-F as well.**

The above detailed discussion on the *meso*-selectivity of PDHG-F has been added in the manuscript and the Supplementary Information. The *de* of *meso*-configuration for PDHG-F is corrected in the manuscript and Supplementary Information.

3. While this is a good idea and the chemistry could have immense practical value, it may also be considered a logical next step from work that is already published. I really enjoyed reading this paper, but nevertheless, it **lacks the high level of novelty** that is required for a journal such as Nature Communications.

Thank you for your comments on this work. This work has been corrected and improved according to your suggestions. For the novelty, **this work initially shows the kinetically stereoselective gridization and polymerization to construct the steric pi-nanoarchitecture units (DHG derivatives) and the nanopolymers (PDHGs) that are different from covalent organic frameworks (Science, 2017, 355, 923-931), graphene derivatives (Nat. Chem. 6, 126-132) and other popular 2D-dimensional structures.** It is noted that constructing the stereoregular 3D nanoarchitectures is much more rigorous than the construction of the planar covalent nano-architectures because the synthesis of 3D

nanoarchitectures **additionally** requires the precise stereocontrol of the chiral centers (at the sp^3 -carbon atoms), which potentially influences the crystallinity, optical properties, microscopic dynamics, drug activities as well as the ionic transfer tunnels (analogous to the tacticity of polymer chains). These properties play a key role in the organic electronics, sensors, molecular machine shell, biomedicine and ionic devices. On the other hand, we can image that if the Friedel-Crafts gridization is in the absence of specific stereocontrol, the thermodynamic equilibrium that efficiently generates massive byproducts (such as macro-ring homologs and linear oligomers) will be dominant. In this case, the isolation will be rather difficult. What's more, the thermodynamic macro-ring homologs (even in less than 1% amount) in the polygridization lead to forming the insoluble cross-linking polymers rather than soluble polygrids during the polygridization process, which is unfavorable to the solution processing technology that is crucial to organic electronic materials. Meanwhile, the architectural regularity will be randomly destroyed and thus the nano-architectures will be unable to characterize, which will lead to failure in researching their corresponding relationships between structures and properties, as well as their potential applications. In addition, some of gridarene isomers, with interesting properties maybe, are kinetic products and thus they are unable to obtain efficiently. For example, according to the theoretical calculation of the Drawing-Hand Gridarene systems (**DHGs**) reported herein, the centrosymmetric meso-configuration (**meso-DHG**) is favorable to construct soluble stereoregular nanopolymers (polygrids as the model) with high rigidity, high structural anisotropy, and distinguished dynamic behaviors of chain collapse, but **meso-DHGs** are the kinetic products with relative unstable structures during the gridization. Thus, **meso-DHGs** are very difficult to effectively afford during the thermodynamic gridization. As a result, **developing the meso-selective gridization that overcomes the thermodynamic obstacle is more significant for the construction of the stereoregular nanorchitectures.** Toward this, we controlled the dimer packing modes (in centrosymmetric manner) of substrates via synergized hydrogen-bond attraction and cationic repulsion to effectively produce meso-configurational gridarene units and even polygrids in-situ. Further, **this polygridization is the model polymerization of the nano-linkage that would potentially create complex covalent nano-architectures (involving ring-chain-iterated meta-polymers).** It is noted that such synthetic technology has been undeveloped to date since the uncontrollable branching pathways exist in the process of the multiple bond linkage (at the chain-ends).

For the polygridization, we can consider such polygridization as the inverse process of the ring-opening polymerization. Though the stereoselective mechanism (chain-end or site-end mechanism) of the ring-opening polymerization has been deeply studied especially for Lactide derivatives to obtain stereoregular polymers with excellent performances such as high thermal stability, high melting points, high crystallinity and high mechanical strength (Nat. commun. 2018, 9, 1559; ACS Macro Lett. 2018, 7, 624–628), the inverse polymerization process (such as the polygridization) especially in terms of the stereoselectivity has not been studied yet. Moreover, the previous mechanisms cannot be applied in such inverse process (including intermolecular linkage and intramolecular gridization steps). Thus, this work firstly affords the stereoselectivity during the inverse process of the ring-opening polymerization. The new mechanism is based on the centrosymmetric packing of monomers in-situ, **without catalyst scaffolds**. Such packing mechanism can be extended in other polycondensations whose stereoselectivity has not

been realized.

In addition, we created the analysis mode of the polygrid structures especially for the GPC spectra of PDHG-F including the calculation of the Mark-Houwink exponent and the number/weight-average of molecular weights (dividing the elution peaks via the extrapolation of the calibration equation). It is noted that the common analysis method of GPC spectra (using polystyrene standard) leads to evident deviations for rigid PDHG-F chains. Other similar rigid polymers such as graphene nanoribbons and ladder polymers suffer from this deviations and have not been perfectly addressed (typical examples: Nat. Chem. 2014, 6, 126-132; Angew. Chem. Int. Ed. 2011, 50, 2540–2543; J. Am. Chem. Soc. 2017, 139, 5801–5807). However, the precise molecular weight calculation of the polygridization is the prerequisite on researching the polymerization kinetics in the field of polymer chemistry, as well as the long-range structures (including the chain flexibility and the conformational transformation) and the aggregation structures (involving chain entanglement) of polygrid chains in the field of polymer physics. As a result, our molecular weight calculation method precisely provides the fundamental structural information of polygrids and enables to extend other kinds of similar topological polymers (especially for the ring-chain alternated polymers), which is fundamental for further research in terms of polymer science and engineering.

In all, this work significantly contributes to the new stereoselective reactions and polymerizations for the construction of new-generational stereoregular nanopolymers, which is probably up to the novelty of the Nature Communications.

Other considerations for the authors:

1. In reactions done with **excess acid**, there are few if any hydrogen bonding sites available for stereoselective control as is suggested in the manuscript (due to protosolvation). A more likely element of stereocontrol is the **pi-stacking** between electron rich aryl groups and the cationic aryl groups.

Thank you for your discussion on the gridization mechanism. The **pi-stacking between electron rich aryl groups and the cationic aryl groups** does play a significant role in the gridization process. However, we deduced that the methoxyl group contributes to the stereocontrol much more than above pi-stacking based on the gridization results between **MC1** and **MC2**.

Fig. R12. The gridizations of MC1 and MC2

We hypothesized that if the donor-acceptor pi-stacking is only a key factor to this

stereocontrol during the gridization process, the substrate **MC2** with the same DAF and carbazole groups would exhibit the similar meso-selectivity like MC1. However, MC2 afforded **meso-DHG2** in 15% yields and **rac-DHG2** in 26% yields (under the same gridization conditions), **reflecting the rac-selectivity**. Consequently, the methoxyl groups are crucial to this meso-selectivity.

Fig. R13. The NOESY spectra of MC1 in CF₃COOH-CDCl₃ solution

Fig. R14. The NOESY spectra of MC2 in CF₃COOH-CDCl₃ solution

In order to detect the differences of substrates' self-assembly in the solution, we compared the **NOESY** of protonated **MC1** solution with the **NOESY** of protonated **MC2** solution (Shown in Fig. R13 and R14 respectively). Based on the standard of the

cross-peak (space correlation of methyl groups to the benzene segments), we observed that the **MC1** displays the stronger cross-peak (1.30, 7.10) which is assigned to the space correlation between the methoxyl groups and the ethyl groups (on the carbazoles). Meanwhile, the weaker cross-peak at the (1.30, 9.30) is assigned to the space correlation between the 3-sites of DAF groups and the ethyl groups (on the carbazoles). Above cross-peaks suggest the anti-parallel packing modes. On the contrary, **these cross-peaks were not observed in the NOESY of protonated MC2 solution! These results indicate that most of MC2 substrates do not undergo such packing modes, which is related to the rac-selectivity and the lower gridization efficiency.** As a result, we deduced that **the methoxyl group plays a key role in supporting such dimer packing.** On this basis, considering the hydrogen-bond acceptor of the oxygen atoms on the methoxyl groups (reference: J. Am. Chem. Soc. 2009, 131, 2629-2637), we deduced that the hydrogen bond on the methoxyl groups is likely to facilitate this donor-acceptor packing. Moreover, no oxygen atoms on the benzene group of methyl-based **MC2** substrates lead to *rac*-selectivity and lower gridization yields, which confirm our deduction. In all, **the attractive interactions on the methoxyl groups stabilize the pi-stacking** and thus realize the meso-selective gridization.

2. In the reactions done with highly acidic conditions, the Friedel-Crafts reactions are possibly reversible. This may provide a basis for the thermodynamic control of product distribution. With **less acidic** conditions, the reversibility of the Friedel-Crafts reactions becomes unlikely.

Thank you for your consideration on the reversibility of the Friedel-Crafts process. Before discussing the reaction reversibility, we stressed that the highly acidic conditions are used to initiate these Friedel-Crafts reactions in superelectrophilic modes (such as at least 30 times of CF₃SO₃H, references: J. Am. Chem. Soc. 2011, 133, 13169-13175; J. Org. Chem. 2017, 82, 6044-6053). **The less acidic conditions do not initiate these reactions.**

Fig. R15. The ^1H NMR spectra of the MC1 gridization systems after reacting specific times. The blue region indicates the signals that belong to rac-configuration while the red region shows the peaks of meso-configuration.

Fig. R16. The reaction results of the MC1 gridization systems including the *de* value of meso-configuration (a) and the yields of corresponding products (b, involving meso-DHG1, rac-DHG1 and MG1).

Inspired by your suggestion (based on the excessive acid conditions), we studied the reaction time effect on the gridization of **MC1** (in Fig. R15, the yield results are shown in Fig. R16). We were happy to find that **1 min affords the highest meso-selectivity (75.6% de) and elongating the reaction time enables to lower the meso-selectivity to 66.7% (1 h), 60% (10 h), 53.8% (24 h), 36.9% (31 h) and even to 19% de (54 h).** By further study (Fig. R16b), elongating the reaction time (1~24 h) leads to increasing the **MG1** yields from 21.5% to 40~50% yields but reducing the **meso-DHG1** yields to 35~40%. Meanwhile, the yield of **rac-DHG1** is almost unchangeable. These results suggest that the **meso-DHG1** are unstable in such acid conditions and transforms to **MG1** in the relative short time. Therefore, **the meso-DHG1 is the kinetic products during the Friedel-Crafts gridization and the supramolecular packing overcomes the thermodynamic limit to support such kinetic stereoselectivity.** Although the yield of **rac-DHG1** is slightly increased to 22% after reacting 54 h, the transformation from MG1 to rac-DHG1 is drastically difficult as well.

Fig. R17 the reaction solution state of PDHG-F after adding the CF₃COOH and CF₃SO₃H

In order to confirm the deduction that the **meso-DHG** units enable to form **MG**, we added the soluble **PDHG-F** in reaction solution (the mixture of DCM, CF₃COOH and CF₃SO₃H) and observed whether the **PDHG-F** enables to form insoluble **CP-F** (cross-linking polymers that derived from MG segments). If the reaction solution do not generate insoluble substance (**CP-F**), such **meso-DHG** will be difficultly transformed or can

be transformed to **rac-DHG** units directly. Otherwise, the **meso-DHG** units can be deduced to form **MG** units easily. Unfortunately, after 15 min, the insoluble **CP-F** were generated and the yield was continuously increased if elongating the reaction time (Shown in **Fig. R17**). **Therefore, the meso-DHG units enable to form MG1 during the reversible process.**

On this basis, **such time effect account for the slightly higher meso-selectivity of polygridization (~80%) because the polygridization is underwent in less than 30 seconds during which the meso-DHG units are not transformed to MG segments (in words, the meso-selectivity was not reduced).**

Similarly, we studied the reaction time effect of **MC2** substrate to testify the supramolecular contributions to the gridization process. In **Fig. R18** and **R19**, the yields of all products (**meso-DHG2**, **rac-DHG2** and **MG2**) are almost maintained or changed negligibly, **indicating no evident reversible process exists during the MC2 gridization process.** Compared with the **MC1** gridization (with evident thermodynamic reversibility), we deduced that the thermodynamic pathways probably dominate such gridization of **MC2** substrates in the beginning stage. Moreover, the yield of kinetic product **meso-DHG2** is the lowest, confirming our deduction. Thus, the supramolecular assembly hardly control the gridization process for methyl-based **MC2** substrates. These results is consistent with the **NOESY** of protonated **MC2**.

Fig. R18. The ¹H NMR spectra of the MC2 gridization systems after reacting specific times. The blue region indicates the signals that belong to rac-configuration while the red region shows the peaks of meso-configuration.

Fig. R19 the reaction results of the MC2 gridization systems including the *de* value of rac-configuration (a) and the yields of corresponding products (b, involving meso-DHG2, rac-DHG2 and MG2).

Fig. R20 the total yields of DHGs and meso-selectivity of different MC substrates.

In terms of above observations of time effect, **we reperformed the gridization and testified the total yields of DHGs and meso-selectivity under the key conditions “in the 1 minutes”** (Fig. R20)! These results are exhibited in the manuscript.

We are very grateful to you because your discussion and suggestions help us to correct these mistakes and add the key conditions (especially for reaction times to suppress the thermodynamic pathways) for the gridization reactions. I really enjoy improving my research abilities to answer your questions and discuss with you. Thank you for your suggestions again.

Reviewer 2:

The manuscript describes a quite interesting system of cyclisation/polymerisation using some unusual monomers. The work is well performed with the products characterised correctly. The manuscript describes the work reasonable well, although the point of the work is unclear and the gridarene nomenclature is ill-defined and appears unnecessary.

Thank you for your comment on the concept of this kind of molecule and polymers. Here I would like to explain the origin and meaning of gridarenes. First, we defined the concept of gridarene as a family of arene-based building blocks that possess grid-shaped configurations with well-defined edges and perpendicular vertexes (arising from sp^3 -carbon atoms). This concept is analogous to the calix[n]arene (*J. Am. Chem. Soc.* 1989, 111, 8192-8200) and pillar[n]arenes (*J. Am. Chem. Soc.* 2008, 130, 5022-5023). However, superior to above both cyclophanes, the gridarene backbones exhibit extendable, installable, scalable and portable potentials for the construction of nanopolymers (*Chin. J. Polym. Sci.* 2017, 25, 87-97; *Chin. J. Chem.* 2020, 38, 103-105) and even covalently hierarchical pi-nanoarchitectures. In fact, the grid concept was firstly mentioned from Lehn in chemistry (*Angew. Chem. Int. Ed.* 1994, 33, 2284-2287) where the square backbones are assembled and fixed by coordination bonds. Compared with their complexed grid chemistry, the gridarenes defined in this paper has similar geometric configurations and extending models. For example, if the coordination compound moieties (dipyridyl groups and a metal cation) are replaced by the spirobifluorene groups, the tic-tac-toe type grids

with eight extension sites (line 23, the first paragraph of the introduction) will be formed. In addition, the grid point (fully covalent) was suggested by one peer in 2014 (*Org. Lett.* 2014, 16, 1748-1751), which is different from the framework that is a substantially two-dimensional or three-dimensional crystallizable polymer rather than a building unit.

Because both “cyclic compounds” and “gridarenes” have similar closed structures, many people easily recognize the “gridarenes” as the common “cyclic compounds”. However, I have to stress that they have different advantages and developing tendencies. For cyclic compounds, most of researches merely concentrate on the utility of the cavities that are usually used to encapsulate small molecules or interlock with other building blocks. By contrast, as the cyclic backbones have no well-defined vertexes and extendable reaction sites, they have difficulty in polymerizing or extending into regular nanoarchitectures. As a result, there are limited topological polymers consisting of cyclic compounds (as the repeating units). For gridarenes, the vertexes of the gridarenes are able to be linearly linked with each other and the extendable reaction sites on the fluorene-like groups (such as the 2-positions of the diazafluorene groups in this paper) are well-controlled. In terms of the orthogonal grid nanostructures and frameworks in Lehn’s work (*Angew. Chem. Int. Ed.* 2004, 43, 3644–3662), the gridarene units are more used to similarly construct rigid and regular nanoarchitectures including one-dimensional nanopolymer chains and two/three-dimensional frameworks. That is the essential differences between “cyclic compounds” and “gridarenes”.

Based on the concept of the “gridarenes”, the reactions which afford gridarene products are defined as the “gridization” (*Tetrahedron*, 2018, 74, 5833-5838; *Eur. J. Org. Chem.* 2018, 7009-7016), as analogous to the cyclization that gives cyclic compounds, catenation that affords catenane-based building blocks (*Chem. Rev.* 2009, 109, 6024–6046) as well as the ladderization that generates ladder-like polymers (*Chem. Sci.*, 2017,8, 2503–2521).

The manuscript would benefit from telling the reader why this work was carried out and what will be done with the results now they have been obtained. These should be added to any revised manuscript

Thank you for the suggestions. The aim of this work is to provide a stereoselective gridization to construct gridarenes (DHG) and the one-dimensional polygridizations that are different from covalent organic frameworks (*Science*, 2017, 355, 923-931), graphene nanoribbons (*Nat. Chem.* 6, 126-132) and other two-dimensional materials. In our previous work, we found the synthetic method of nanogrid units that can be extensively applied in organic electronics (such as light-emitting diodes and transistor memory). Unfortunately, these synthetic method is not so good that the sizes of grid products cannot be well-controlled. More severely, the stereoselectivity in above gridization was poor and negligible, which generates a series of isomers that are difficult to be isolated. If these synthetic technologies in the absence of stereocontrol are applied to construct pi-nanoarchitectures and nanopolymers, the regularity or tacticity of these nanostructures are completely destroyed, which will be unable to characterize (on their molecular structures) and research the structure-property relationships, as well as their application and processing. Furthermore, the chirality of the building blocks can effectively influence

the crystallinity (Science, 2019, 363, 1439-1443), photo physics (Chem, 2019, 5, 2470-2483) and drug activity (J. Am. Chem. Soc. 2017, 139, 4294-4297). Meanwhile, the stereoselectivity of the polygridization is a crucial point to obtain regular chain configurations that affect their chain conformations, entanglements and organizations followed by film morphology and mechanical properties and so on. Toward this, developing the stereoselective gridization and polygridization (discovered in this work) is a significant improvement not only for the synthetic technologies but also for gridarene-based organic optoelectronic materials, as well as for the covalent nano-technology.

More importantly, such synthetic chemistry is an effective tool to construct a series of grid-typed nanopolymers with special tacticity. Nanopolymers are nanomolecular polymers (indicating the repeating units should be nanoscale, J. Am. Chem. Soc. 1997, 119, 6440-6441), which comes from poly(macromonomer)s (Macromolecules 1994, 27, 1662-1664). The previous reports exhibit that they have ultrarigid mainchains with >100 nm of Kuhn length (Macromol. Rapid Commun. 1994, 15, 279-284), which is approximate to carbon nanotubes. Meanwhile, different from inorganic nanorod and nanowires, they can be soluble and processed by solution technology (such property is similar to polymer materials). To date, many famous nanoscale building blocks including porphyrins (Science, 2001, 293, 79-82), fullerenes (Science, 1993, 259, 955-957) and graphene nanoribbons (Nat. Chem. 2014, 6, 126-132) have been linked into nanopolymer models, which have been applied in molecular antennas (Chem. Soc. Rev., 2015, 44, 943-969), channels of charge transfer (J. Am. Chem. Soc. 2001, 123, 6714-6715) and semipermeable membranes (Macromol. Rapid Commun. 2004, 25, 1674-1678). Distinguished from above examples, the grid-typed nanopolymers exhibit wide band-gaps of semiconductors (ACS Appl. Polym. Mater. 2019, 1, 2441-2449; Chin. J. Polym. Sci. 2017, 35, 87-97). However, as was mentioned above, the gridarene units have at least two chiral centers which evidently influence the chain tacticity of nanopolymers, as tied to their characterization, structure-properties relationships and applications. As a result, this work will provide the synthetic stereocontrol method for the construction of stereoregular nanopolymers potential in the application of organic wide-bandgap semiconductors.

In terms of the meso-selective results on the polygrid backbones, we want to find out the effect of meso-configurational repeating units on the nanopolymer chain dynamics via molecular dynamic simulation. Among of them, considering that the chain collapse always serves as the structural defects of polymer semiconductors (Nature. 2000, 405, 1030-1033) which are sensitive to the optoelectronic properties (J. Am. Chem. Soc. 2000, 122, 2244-2251), we simulated the chain collapse of these polygrids via atomistic calculation. **It is noted that these nanopolymers are not collapsed in common solvents such as DCM, tetrahydrofuran and CHCl₃. However, such simulation provides polygrid chains with the vacuum environment (not real models) analogous to poor solvent (possessing extremely low viscosity) that powerfully supports the chain collapse.** The structures are shown below (Fig. R21), including **meso-PDHG-F** (100% meso-configuration on the main-chain) and **rac-PDHG-F** (shown in Fig. R21).

Fig. R21 the constructed structures that will perform the simulation of chain collapse

Fig. R22. The molecular dynamic simulation of the single PDHG-F (including *meso*-PDHG-F and *rac*-PDHG-F) chain collapse under the vacuum conditions (298 K, 0 GPa). The conformations of expanded chains (EC stage), sausage-like chains (SC stage) and collapsed chains (CC stage) during the collapsed process are shown in a), where the DP was set as 13. The chain conformations before performing collapse is obtained by geometry optimization. In order to research the anisotropy of the collapsed conformations, we used the ellipsoid models that covers the main-chain backbones. The eccentricity of the ellipsoid model (e) is used to quantitatively evaluate the conformational anisotropy in

collapsed state ($e = 1$ for the 100% anisotropy and $e = 0$ for the perfect isotropy), which is depicted in b) during the collapsed procedure. The including covalent energy (Valence) is shown in c).

Inspired by the analogous tacticity in polyesters (Nat. Commun. 2018, 9, 1559) and poly(vinylether)s (Science, 2019, **363**, 1439-1443), we study the *meso*-configurational effect on the chain collapse via atomistic simulation. During the dynamics (NVT ensemble, 298 K), **meso-PDHG-F** performs the conformational transition from the linear rod (EC stage) to the folded sausage (SC stage) followed by the collapsed rod (cluster stage, in **Fig. 5a**). The eccentricity of its ellipsoid model is decreased from 0.99 to 0.91 and is equilibrated at 0.90~0.91 (**Fig. 5b**), showing the high structural anisotropy. By contrast, **rac-PDHG-F** carries out the conformational transition (**Fig. 5a**) from the helical backbone (EC stage) to the cyclized motion (SC stage) and the toroid main-chain (cluster stage). The eccentricity of its ellipsoid model is drastically fluctuated (0.96~0.11) and is equilibrated at 0.47~0.57 with low anisotropy. Both collapsed main-chains, distinguished from traditional globule or sphere manners, were predicted merely with coarse-grained models rather than atomistic simulation (Nature, 2000, 405, 1030-1033). During the collapsed process, **meso-PDHG-F** undergoes the rotation of carbazole groups toward syn-type DHG-edges to crumple the chains, while the rotation between C_2 -symmetric *meso*-DHG units and fluorene groups maintains the linear backbones. Conversely, initiating the **rac-PDHG-F** collapse only originates from the rotation between the folded **rac-DHG** units and fluorene groups. Consequently, the collapsed **meso-PDHG-F** possesses higher average covalent energy (7.3 kcal/mol per repeating units) than that of collapsed **rac-PDHG-F** (**Fig. 5c**), which suggests that the *meso*-selective **PDHG-F** displays higher resistance of chain collapse. The above observations are majorly confirmed by other polygrids with various DP (in the Supplementary Information).

It is noted that the present research on the polymer tacticity is limited to the crystallinity, the glass transition, the melting point as well as the mechanical properties (Science, 2019, **363**, 1439-1443), in terms of flexible chains. By contrast, the research of the tacticity effect on the chain collapse, for the rigid rod-like nanopolymers, is rare. Therefore, these results initially reveal the collapsed tacticity of the nanopolymers, which will help us to find out the interesting microscopic dynamics that can be extended into covalent nano-architectures.

These discussion has been supplemented in the manuscript and supplementary information. Thank you for your suggestion again.

REVIEWERS' COMMENTS:

Reviewer #1 (Remarks to the Author):

This revised manuscript is significantly improved over the initial submission. Moreover, the authors have provided convincing evidence for the stereoselective formation of the polygridization products (PDHGs).

Regarding the authors' responses to reviewer comments, they did an admirable job answering points and questions. On page 20-23, the authors use NOESY experiments to support their idea that hydrogen bonding interactions with the methoxy group are important in packing interactions. Maybe. But the NOESY experimental conditions may not be a good model for the conditions which lead to the PDHG products. The key step in grid formation must involve ionization to a superelectrophilic carbocation, one which an aza-fluorenyl cation is substituted by an aryl group (p-tolyl or p-methoxyphenyl). Recent studies have shown that these types of highly ionized structures involve large-scale migration or delocalization of the carbocation charge into the aryl group (J. Am. Chem. Soc. 2011, 133, 13169). In particular, the p-methoxyphenyl group would largely exist in the rigid, quinonemethide-type of structure (a carboxonium ion with little or no free bond rotation). The p-tolyl group is not expected to have this type of charge delocalization and this could manifest itself in higher electrophilic reactivity compared to the p-methoxyphenyl and free rotation about the C-C bond (between aza-fluorenyl group and p-tolyl group). These effects may influence the stereochemical outcome of ring-formation.

The authors may also wish to keep in mind that during the course of this chemistry, water is being released and this will significantly effect the acidity of the media.

As mentioned in the previous review of this submission, the chemistry does have the novelty to make it stand out. This is an interesting chemical systems and macromolecular structure. This is the type of paper that has something for almost everyone - material science, polymer chemistry, mechanistic organic chemistry, synthetic organic chemistry, nano chemistry, and so on. In the opinion of this reviewer, the paper is very compelling and readable.

Note for answer the questions from reviewers

Reviewer 1:

1. This revised manuscript is significantly improved over the initial submission. Moreover, the authors have provided convincing evidence for the stereoselective formation of the polygridization products (PDHG).

Regarding the authors' responses to reviewer comments, they did an admirable job answering points and questions. On page 20-23, the authors use NOESY experiments to support their idea that hydrogen bonding interactions with the methoxy group are important in packing interactions. Maybe. But the NOESY experimental conditions may not be a good model for the conditions which lead to the PDHG products.

Thank you for your suggestion and discussion. For the NOESY experiments, the characterized condition is CF_3COOH and CDCl_3 mixed solvents (the volume ratio of CF_3COOH and CSCl_3 is 1:3). Under such condition, the substrates **MC1** (with p-methoxyphenyl group), **MC2** (with p-tolyl group) and **DC-F** are fully protonated but the carbocationic species are not generated. When adding $\text{CF}_3\text{SO}_3\text{H}$, these substrates are transformed into superelectronophiles that quickly undergo gridization process (**in 10 seconds**). Thus, it is extremely difficult to directly observe the molecular packing modes of superelectronophiles in such a short time. However, **the superelectronophilic stacking should be closely related to the molecular packing of protonated substrates with the same supramolecular functional groups** including diazafluorenes (in protonated state), methoxyl groups and carbazole moieties. In this regard, we considered the compromising method that uses the NOESY spectra of protonated **MC** substrates (without $\text{CF}_3\text{SO}_3\text{H}$) to study their plausible molecular packing, which serves as the basis for the deduction of superelectrophilic stacking. As is shown in **Fig. R1a**, the protonated **MC1** displays the stronger cross-peak (1.30, 7.10) which is assigned to the space correlation between the methoxyl groups and the ethyl groups (on the carbazoles). Meanwhile, the weaker cross-peak at the (1.30, 9.30) is assigned to the space correlation between the 3-sites of diazafluorene groups and the ethyl groups (on the carbazoles). **These cross-peaks suggest that the protonated MC1 has already performed anti-parallel packing even in the absence of carbocation.** Correspondingly, the gridization of **MC1** substrates exhibits high *meso*-selectivity, evident time effect and high gridization yields (**Fig. R1b**). In contrast, the protonated **MC2** does not possess these cross-peaks (**Fig. R1c**), indicating the much weaker molecular packing. Interestingly, the gridization of **MC2** exhibits the *rac*-selectivity, negligible time effect and lower gridization efficiency (**Fig. R1d**). As a result, **such comparison reflects the potential relationship between the molecular packing and the gridization pathways.** Although the NOESY condition may not be the a good model for the conditions generating DHG products, the NOESY results do uncover the molecular packing of MC substrates that supportively inspired us to discover the gridization mechanism.

Fig. R1. The NOESY spectra of protonated MC1 (a) and MC2 (c). The stereoselectivity along with time effect are shown as well in (b) for MC1 reaction systems and in (d) for MC2 reaction systems as well. The gridization conditions of MC1 are the same as that of MC2 (DCM solvent, CF₃COOH as the acid additive, CF₃SO₃H as the acid catalyst, 10~15 mM of the substrate concentration).

The key step in grid formation must involve ionization to a superelectrophilic carbocation, one which an aza-fluorenyl cation is substituted by an aryl group (p-tolyl or p-methoxyphenyl). Recent studies have shown that these types of highly ionized structures involve large-scale migration or delocalization of the carbocation charge into the aryl group (J. Am. Chem. Soc. 2011, 133, 13169). In particular, the p-methoxyphenyl group would largely exist in the rigid, quinonemethide-type of structure (a carboxonium ion with little or no free bond rotation). The p-tolyl group is not expected to have this type of charge delocalization and this could manifest itself in higher electrophilic reactivity compared to the p-methoxyphenyl and free rotation about the C-C bond (between aza-fluorenyl group and p-tolyl group). These effects may influence the stereochemical outcome of ring-formation.

Fig. R2. The plausible gridization mechanism of MC1 that involves the carbocationic delocalization step. The reaction pathways are shown in (a). The ball-and-stick models of the likely intermediate V, VI, VII and VIII are depicted in (b).

Thank you very much for your valuable suggestion. **Your discussions are reasonable, we accepted your idea of charge delocalization** (J. Am. Chem. Soc. 2011, 133, 13169–13175) in the mechanism of gridization. Indeed, the diazafluorene-based superelectrophiles perform the charge delocalization as well (Org. Lett. 2016, 18, 6220-6223; Arkivoc 2018, part ii, 215-232). Moreover, your opinions can be introduced into the molecular packing mechanism without any contradiction. The detailed analysis is

shown below:

The tricationic species **II** undergo the migration of the carbocation (**III**) to obtain the quinonemethide-type intermediates **IV** that performs structural planarization between the diazafluorene group and the methoxybenzyl moiety (fixed by double bonds, **Fig. R2a**, this process has been supplemented in the **Supplementary Fig. 23**). The superelectrophilic quinonemethide-type examples have been exhibited in the literature: Chem. Eur. J. 2008, 14, 2004 – 2015. In this case, **the cationic methoxybenzyl group probably interacts with the electron-donating carbazole moiety (via pi-pi stacking) and powerfully strengthens the centrosymmetric molecular packing V, which is favorable to meso-selectivity. By contrast, the rotation of methoxybenzyl groups (VI) is unfavorable to the pi-pi stacking between two MC1 backbones (Fig. R2b).** For the asymmetric packing mode (**VIII**) that generates *rac*-DHGs, the quinonemethide-type intermediates **IV** likely make negligible contributions to enhancing molecular packing **VII**, which is unable to efficiently improve *rac*-selectivity. Hence, **the quinonemethide-type structures are seemingly favorable to centrosymmetric packing that enhances the meso-selectivity**, which is confirmed by the *rac*-selectivity of tolyl-based **MC2** substrates without quinonemethide-type species.

Fig. R3. The gridization of F-MC substrates and the ¹H NMR spectra of the reaction systems

Distinguished from the regioselectivity that has been detailed in the literature (J. Am. Chem. Soc. 2011, 133, 13169–13175), the formation of C-C bond (during the gridization stage) still occurs at the 9-position on diazafluorene groups, which overcomes the steric hindrance of congested tetraarylmethane. Thus, it is rather difficult to directly demonstrate such cationic delocalization. Nevertheless, **we designed**

the fluorene-based mono-crescent substrates F-MC and performed the similar gridization (**Fig. R3**). **Distinct from diazafluorene-based substrates, the mono-cationic species of the fluorene backbones cannot transform into superelectrophiles.** Hence, in the absence of repulsive driving force, the charge delocalization toward the methoxybenzyl moiety will not occur. We hypothesized that if charge delocalization does not influence the stereoselectivity, the substrate **F-MC** will still afford dominant *meso*-selective products. Under the CF_3COOH acid conditions (in 1 minute), we found that **the substrate F-MC exhibits the 1:1.6 ratio of *meso*/*rac*-FDHG (corresponding to 23.1% *de* of *rac*-configuration, Fig. R3)**, rather than dominant *meso*-selectivity. Therefore, the above results indicate that the charge delocalization is an essential contributions to the *meso*-selectivity.

We are happy to say that the charge delocalization does help us understand the stereoselective gridization further during the molecular packing process, although the direct evidence is very difficult to provide.

The process of charge delocalization has been complemented in the manuscript and Supplementary Information, but the gridization results of **F-MC** are summarized in another paper that will be completed.

2. The authors may also wish to keep in mind that during the course of this chemistry, water is being released and this will significantly affect the acidity of the media.

Thank you for reminding us about the released water. Under these gridization conditions, **the molar ratio of the water (released from MC substrates): CF_3COOH : $\text{CF}_3\text{SO}_3\text{H}$ is about 1:300:75.** The 0.3% amount of water should not obviously lower the acidity of the media that is maintained by highly excessive equivalents of strong acids. The good gridization yields and *meso*-selectivity confirm this point. By contrast, if we **directly add massive water in the reaction systems of DC-F (without the termination via MC1)**, the insoluble cross-linking polymers will be majorly formed. Thus, the extremely low amount of released water is negligible during the optimized gridization and polygridization. Anyway, we are grateful to your reminding about the released water.

3. As mentioned in the previous review of this submission, the chemistry does have the novelty to make it stand out. This is an interesting chemical systems and macromolecular structure. This is the type of paper that has something for almost everyone - material science, polymer chemistry, mechanistic organic chemistry, synthetic organic chemistry, nano chemistry, and so on. In the opinion of this reviewer, the paper is very compelling and readable.

Thank you for your comments. The suggestions especially for charge delocalization have been considered and supplemented in the manuscript and Supplementary Information. We are happy to get further insight into the gridization mechanism from your viewpoints.

Reviewers' comments:

Reviewer #1 (Remarks to the Author):

The authors have adequately addressed questions and concerns raised from this reviewer during previous evaluations of this manuscript.

The merits of this submission have been previously outlined.